# EphrinB2 knockdown in cervical spinal cord preserves diaphragm innervation in a mutant SOD1 mouse model of ALS

Mark W Urban[1], Brittany A Charsar[1], Nicolette M Heinsinger[1], Shashirekha S Markandaiah[2], Lindsay Sprimont[1], Wei Zhou[1], Eric V Brown[1], Nathan T Henderson[1], Samantha J Thomas[1], Biswarup Ghosh[1], Rachel E Cain[1], Davide Trotti[2], Piera Pasinelli[2], Megan C Wright[3], Matthew B Dalva[1,4]*, Angelo C Lepore[1]*

[1]Department of Neuroscience, Jefferson Synaptic Biology Center, Vickie and Jack Farber Institute for Neuroscience, Sidney Kimmel Medical College at Thomas Jefferson University, Philadelphia, United States; [2]Jefferson Weinberg ALS Center, Department of Neuroscience, Vickie and Jack Farber Institute for Neuroscience, Thomas Jefferson University, Philadelphia, United States; [3]Department of Biology, Arcadia University, Glenside, United States; [4]Department of Cell and Molecular Biology, Tulane Brain Institute, Tulane University, New Orleans, United States

*For correspondence:
matthew.dalva@jefferson.edu
(MBD);
Angelo.Lepore@jefferson.edu
(ACL)

**Competing interest:** The authors declare that no competing interests exist.

**Abstract** Amyotrophic lateral sclerosis (ALS) is a neurodegenerative disease characterized by motor neuron loss. Importantly, non-neuronal cell types such as astrocytes also play significant roles in disease pathogenesis. However, mechanisms of astrocyte contribution to ALS remain incompletely understood. Astrocyte involvement suggests that transcellular signaling may play a role in disease. We examined contribution of transmembrane signaling molecule ephrinB2 to ALS pathogenesis, in particular its role in driving motor neuron damage by spinal cord astrocytes. In symptomatic SOD1$^{G93A}$ mice (a well-established ALS model), ephrinB2 expression was dramatically increased in ventral horn astrocytes. Reducing ephrinB2 in the cervical spinal cord ventral horn via viral-mediated shRNA delivery reduced motor neuron loss and preserved respiratory function by maintaining phrenic motor neuron innervation of diaphragm. EphrinB2 expression was also elevated in human ALS spinal cord. These findings implicate ephrinB2 upregulation as both a transcellular signaling mechanism in mutant SOD1-associated ALS and a promising therapeutic target.

## eLife assessment

This is a **valuable** study of Eph-Ephrin signaling mechanisms generating pathological changes in amyotropic lateral sclerosis. There are exciting findings bearing on the role of glial cells in this pathology. The study emerges with **solid** evidence for a novel astrocyte-mediated mechanism for disease propagation. It may help identify potential therapeutic targets.

## Introduction

Astrocytes are glial cells that play critical roles in CNS function and dysfunction, including in neurodegenerative diseases such as amyotrophic lateral sclerosis (ALS; *Pekny and Nilsson, 2005*). In ALS, loss of upper motor neurons (MNs) in the brain and lower MNs of spinal cord and brainstem results in progressive muscle paralysis and ultimately in death, usually in only 2–5 years after diagnosis (*Bruijn et al., 2004*). The majority of ALS cases are sporadic, while 10% are of the familial form; these familial

cases are linked to a variety of genes such as Cu/Zn superoxide dismutase 1 (SOD1; *Rosen, 1993*), TAR DNA-binding protein 43 (TDP-43) (*Mackenzie et al., 2007*), C9orf72 hexanucleotide repeat expansion (*DeJesus-Hernandez et al., 2011*; *Renton et al., 2011*) and others.

While ALS is characterized primarily by MN degeneration, studies with human ALS tissue and experiments in animal and in vitro models of ALS demonstrate that cellular abnormalities are not limited to MNs (*Ilieva et al., 2009*). In particular, non-neuronal cell types such as astrocytes play significant roles in disease pathogenesis. Findings suggest that transcellular signaling between astrocytes and MNs may represent an important regulatory node for MN survival and disease progression in ALS (*Yamanaka and Komine, 2018*). However, the mechanistic contributions of astrocytes to ALS remain incompletely understood, hampering development of effective therapies for targeting this cell population and for treating the disease.

One family of proteins linked to both transcellular signaling and ALS are the erythropoietin-producing human hepatocellular receptors (Ephs) and the Eph receptor-interacting proteins (ephrins) (*Schmidt et al., 2009*). Ephs are transmembrane signaling molecules and the largest known family of receptor tyrosine kinases in the mammalian genome. Ephs bind to and are activated by ephrins, which are either glycosylphosphatidylinositol-linked (ephrin-As) or are transmembrane proteins (ephrin-Bs) also capable of signaling (*Klein, 2009*; *Pasquale, 2008*). Eph-ephrin transcellular signaling regulates many events in the developing and mature nervous system that are mediated by cell contact dependent mechanisms, including: dendritic spine formation, dendritic filopodia-dependent synaptogenesis, axon guidance, control of synapse maintenance and density, and synaptic localization of glutamate receptor subunits (*Klein, 2009*; *Pasquale, 2008*). Eph-ephrin signaling is an important mediator of signaling between neurons and non-neuronal cells in the nervous system. Neuronal EphA binding to glial ephrin plays an important role in the morphogenesis of dendritic spines (*Murai et al., 2003*). In the peripheral nervous system, axons expressing EphA are guided to their correct target via ephrin-Bs expressed in the limb bud. Moreover, during development EphA4 is expressed by MNs undergoing programed cell death (*Ohta et al., 1996*), while blockade of EphA4 signaling can limit cell death in models of stroke (*Kandouz, 2018*; *Li et al., 2012*). Thus, Eph-ephrin transcellular signaling is a potent modulator of neuronal function and survival.

In the mature CNS, dysregulation of Eph and ephrin signaling has been linked to a number of neurodegenerative diseases, including ALS. Expression of EphA4 in MNs significantly contributes to MN degeneration and overall disease pathogenesis in both rodent and zebrafish animal models of ALS (*Van Hoecke et al., 2012*), while reduction of ephrinA5 worsens disease outcome in an ALS mouse model (*Rué et al., 2019a*). Furthermore, increased EphA4 expression levels and EphA4 signaling capacity correlate with the degree of human ALS disease severity (*Van Hoecke et al., 2012*). While antisense oligonucleotide (*Ling et al., 2018*) or ubiquitous genetic knockdown (*Rué et al., 2019b*) of EphA4 in ALS mouse models does not affect disease phenotype, inhibition of EphA4 signaling using EphA4-Fc partially preserves motor function and MN-specific genetic knockdown delays symptomatic onset and protect MNs in ALS mice (*Zhao et al., 2018*).

In search of new targets to modulate Eph-ephrin signaling, we chose to explore ephrinB2 given previous work showing its role in astrocytes in the disease pathology of other neurological conditions such as traumatic spinal cord injury (*Bundesen et al., 2003*; *Fabes et al., 2006*). We find that ephrinB2 expression in ventral horn astrocytes increases with disease progression in the SOD1[G93A] mouse model of ALS. Patients ultimately succumb to ALS because of respiratory compromise due in part to loss of respiratory phrenic MNs (PhMNs) that innervate the diaphragm (*Mitsumoto et al., 1998*. We therefore tested in the current study viral vector-based small hairpin RNAs (shRNA) knockdown of ephrinB2 *McClelland et al., 2009*) in the ventral horn of SOD1[G93A] cervical spinal cord (*Lepore et al., 2008b*). We evaluated in vivo effects on key outcomes associated with human ALS, including protection of cervical MNs (*Nicaise et al., 2013*; *Nicaise et al., 2012a*; *Nicaise et al., 2012b*), maintenance of diaphragm function and innervation by PhMNs (*Lepore et al., 2008b*; *Lepore et al., 2011b*; *Lepore et al., 2010*), and overall phenotypic disease extension (*Lepore et al., 2007*). Collectively, data from our study provides insights into both disease mechanisms governing MN loss in mutant SOD1 ALS and a potential therapeutic target.

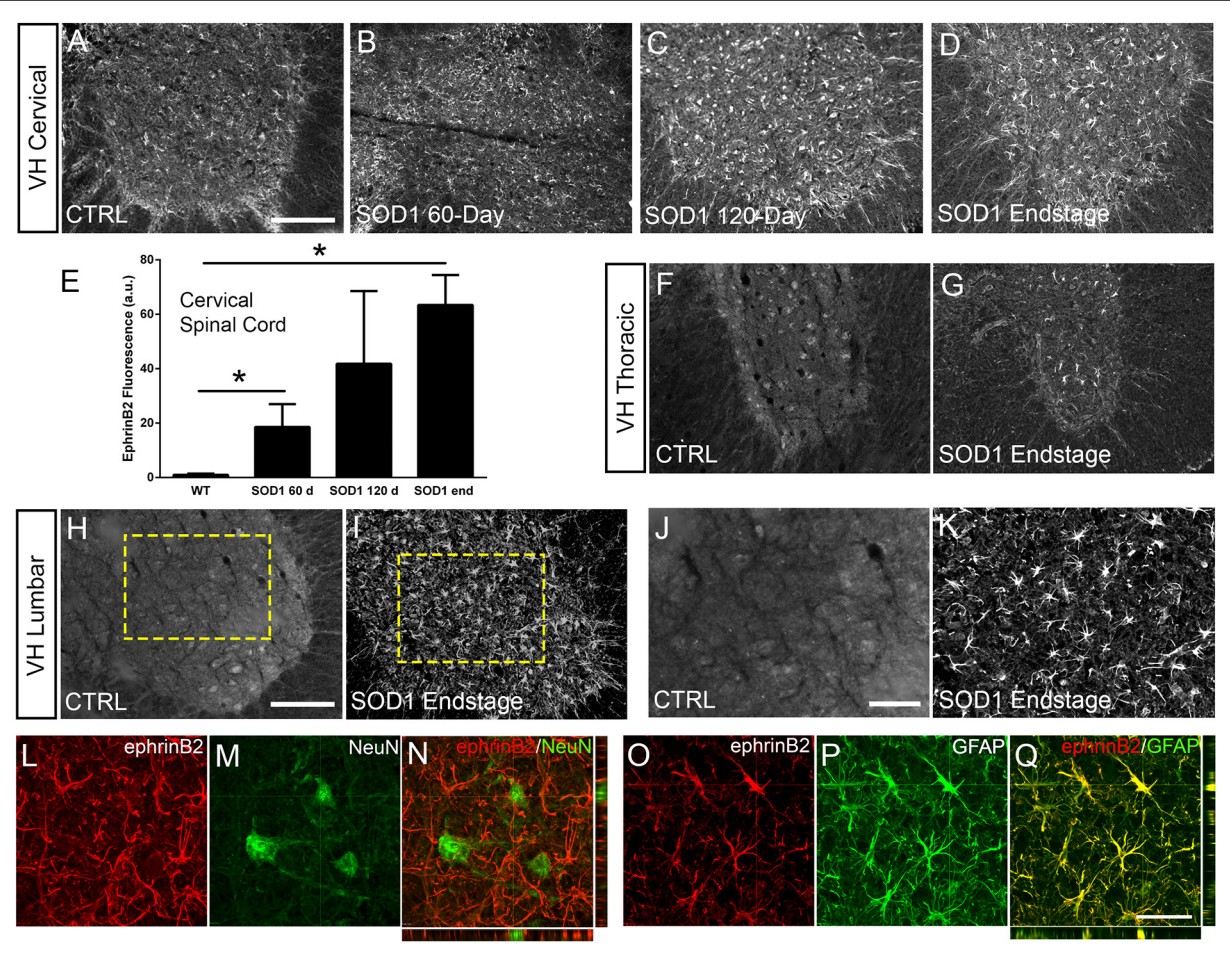

**Figure 1.** EphrinB2 expression was increased in ventral horn astrocytes. SOD1[G93A] mouse ventral horn cervical spinal cord tissue immuinostained for ephrinB2 at 60 days (**b**), 120 days (**c**), endstage (**d**), and WT mouse age-matched control (**a**), scale bar: 200 μm. Quantification of ephrinB2 expression within the ventral horn shows a progressive increase in expression over time compared to WT controls (**e**). Endstage ephrinB2 expression in the thoracic (**f, g**) and lumbar (**h–k**) regions; scale bar: 200 μm, 100 μm, respectively. Endstage SOD1[G93A] mouse cervical spinal cord tissue co-immuostained for ephrinB2 (**l, n, o, q**) and neuronal and astrocyte lineage-specific markers NeuN (**m–n**) and GFAP (**p–q**), respectively; scale bar: 30 μm. Analysis in panels A-K: n=3–4 mice per genotype and per time point; 1–2 females and 2 males per condition. Analysis in panels L-O: n=3 mice per genotype and per time point; 1 female and 2 males per condition.

The online version of this article includes the following source data for figure 1:

**Source data 1.** File contains the raw data for *Figure 1*, panel E.

## Results

### Increase in ventral horn ephrinB2 with disease progression

Eph receptor signaling has been implicated in ALS (*Van Hoecke et al., 2012*; *Tsuda et al., 2008*); however, the involvement of specific ephrin ligands in disease remains unresolved. To begin to address this question, we assessed ephrinB2 expression over the course of disease in SOD1[G93A] mice at pre-symptomatic (60 days), symptomatic (120 days) and endstage time points using ephrinB2 immuno-histochemistry (IHC). We focused in particular on the cervical ventral horn, as it is the location of PhMNs critical to maintaining diaphragm function (*Warren and Alilain, 2014*). In age-matched wild-type (WT) littermates, ephrinB2 was expressed at relatively low levels. Compared to WT controls (*Figure 1a*), there was pronounced up-regulation of ephrinB2 in ventral horn even at the late pre-symptomatic (60 day) time point (*Figure 1b*). EphrinB2 expression dramatically increased over disease course in SOD1[G93A] mice as seen at the symptomatic (120 day) time point (*Figure 1c*) and at disease endstage (*Figure 1d*). Quantification of ephrinB2 expression in the cervical spinal cord ventral horn showed an increase in expression at 60 days (18.58±8.42 a.u. fold increase; WT vs. 60d: p=0.048),

120 days (41.83±26.67 a.u. fold increase; WT vs. 120d: p=0.21; 60d vs. 120d: p=0.32) and endstage (63.42±10.99 a.u. fold increase; WT vs. endstage: p=0.015; 60d vs. endstage: p=0.017; 120d vs. endstage: p=0.3) compared to WT age-matched controls (1.00±0.40 a.u.) (*Figure 1e*; n=3–4 mice per group). Compared to WT control, ephrinB2 expression was also significantly increased at endstage in the thoracic (*Figure 1f, g*) and lumbar (*Figure 1h–k*) ventral horn. Increases in ephrinB2 expression were localized to spinal cord gray matter. Higher magnification imaging from lumbar spinal cord revealed that the vast majority of ephrinB2-expressing cells within the ventral horn displayed an astrocyte-like morphology (*Figure 1j, k*). These data indicate that ephrinB2 expression is upregulated in SOD1$^{G93A}$ mice and suggest that increases in ephrinB2 expression might be localized to glia.

## EphrinB2 was upregulated in ventral horn astrocytes

We next asked whether ephrinB2 was expressed in astrocytes. Expression of ephrinB2 was determined in neurons and astrocytes within ventral horn at disease endstage using double-IHC for ephrinB2 along with lineage-specific antibodies for reactive astrocytes (GFAP: glial fibrillary acidic protein) and for neurons (NeuN: neuronal nuclear protein; *Lepore and Fischer, 2005*). EphrinB2 upregulation was localized to GFAP-expressing astrocytes (*Figure 1o–q*) and was not co-localized to NeuN-expressing neurons (*Figure 1l–n*; n=3 mice). Thus, ephrinB2 expression is dramatically and selectively increased in reactive astrocytes of the SOD1$^{G93A}$ mouse spinal cord in areas of MN loss.

## EphrinB2 knockdown in astrocytes of cervical ventral horn

Given that both ALS patients (*Mitsumoto et al., 1998*) and mutant SOD1 rodents (*Lladó et al., 2006*) succumb to disease due in part to diaphragmatic respiratory compromise, we next sought to focally reduce ephrinB2 expression in astrocytes in the region of the spinal cord containing respiratory PhMNs. To begin to test whether the increased expression of ephrinB2 might impact disease progression, we injected 60-day-old SOD1$^{G93A}$ mice with either lentivirus-GFP control vector or lentivirus that transduces an *Efnb2* shRNA expression cassette (*McClelland et al., 2009*). Virus was injected bilaterally into the ventral horn at six sites throughout the C3-C5 region to bilaterally target the region of the spinal cord containing the diaphragmatic respiratory PhMN pool (*Figure 2a–b*; *Lepore et al., 2008b*). We have shown previously that this shRNA construct selectively targets *Efnb2* and that knockdown effects of the shRNA on ephrinB2 levels are rescued by expression of a shRNA-insensitive version of *Efnb2* (*McClelland et al., 2009*).

We first determined whether our knockdown approach efficiently transduced astrocytes and reduced ephrinB2 expression in spinal cord astrocytes. Transverse sections of endstage SOD1$^{G93A}$ mouse cervical spinal cord show robust expression of the GFP reporter bilaterally within ventral horn following intraspinal injection (*Figure 2c*). To evaluate cell lineage of viral transduction, we performed IHC on lenti-GFP transduced spinal cord tissue at disease endstage. The majority of GFP-expressing cells in ventral horn were GFAP+ reactive astrocytes; these GFAP+/GFP+ astrocytes also expressed high levels of ephrinB2, demonstrating that the lentiviral constructs targeted reactive astrocytes that included those with upregulated ephrinB2 expression (*Figure 2j–l*). On the contrary, there was little-to-no co-labeling of the GFP reporter with NeuN+ neurons (*Figure 2d–f*) or cells positive for oligodendrocyte transcription factor 2 (Olig2) (*Figure 2g–i*), demonstrating that the injected viral constructs did not target a large portion of neurons or cells of the oligodendrocyte lineage within the ventral horn. We quantified the percentage of transduced GFP+ cells that co-labeled with GFAP, NeuN or Olig2 and found that the majority of transduced cells were astrocytes (NeuN: 3.54 ± 1.09% labeled cells, n=3; Olig2: 0.18 ± 0.18% labeled cells, n=3; GFAP: 56.53 ± 5.30% labeled cells, n=3 mice; GFAP vs. NeuN: p=0.023; GFAP vs. Olig2: p=0.018) (*Figure 2m*). We next determined whether the lenti-shRNA vector effectively reduced ephrinB2 expression in ventral horn astrocytes. Compared to lenti-GFP control (*Figure 2o–q*), the lenti-shRNA (*Figure 2p–r*) reduced ephrinB2 expression by approximately a factor of 5 (Lenti-GFP: 86.92±22.35 GFP+/ephrinB2+ cells, n=3 mice; Lenti-shRNA: 16.67±1.76 GFP+/ephrinB2+ cells, n=3 mice mice; t-test, p=0.035; *Figure 2n*). Together, these results show that viral transduction was anatomically targeted to the cervical ventral horn, was relatively specific to the astrocyte lineage, and was able to significantly reduce ephrinB2 expression levels within the C3-C5 ventral horn of SOD1$^{G93A}$ mice.

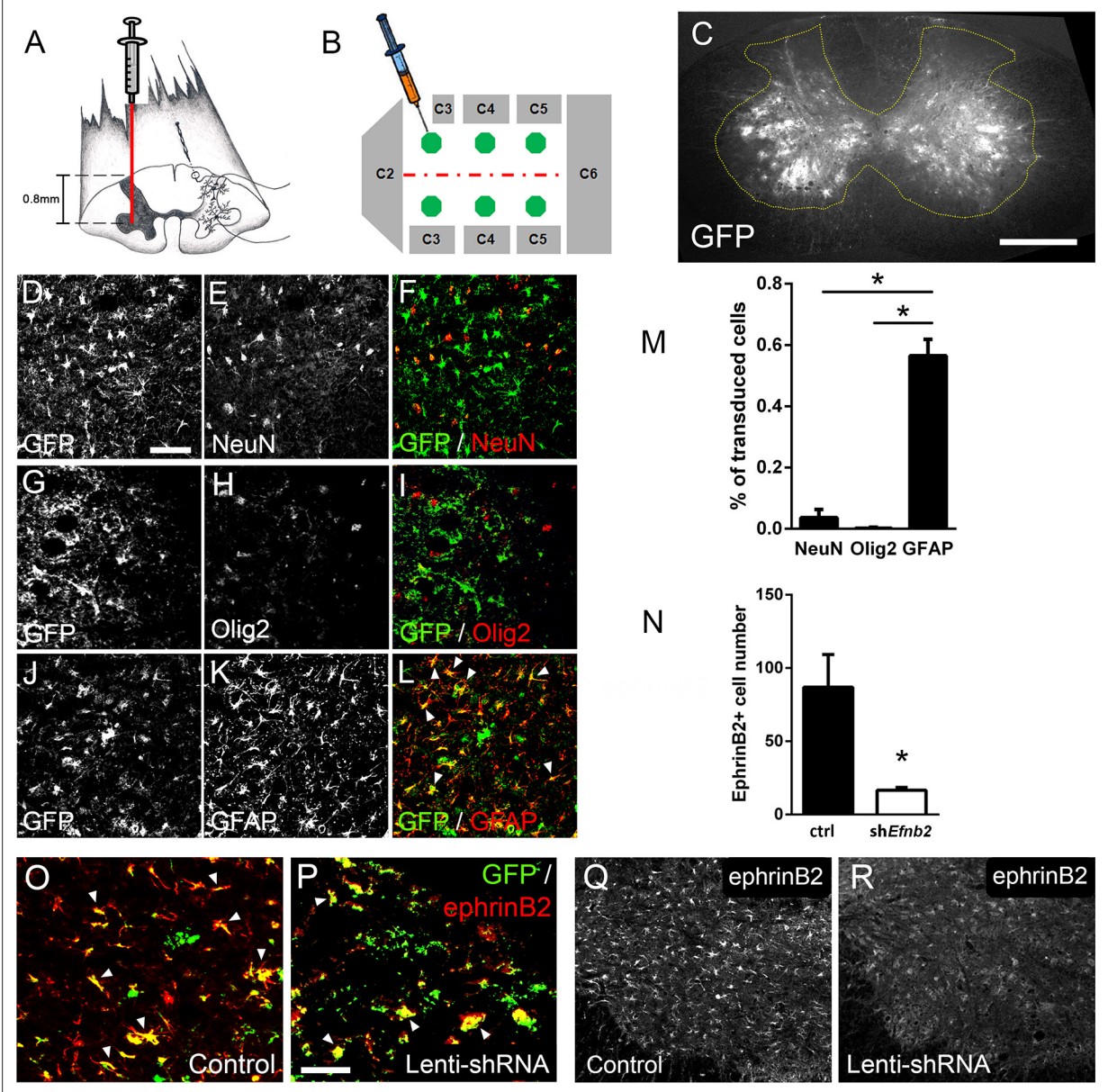

**Figure 2.** Lenti-shRNA injection reduced ephrinB2 expression in cervical spinal cord astrocytes. We injected 60-day-old SOD1[G93A] mice with Lenti-GFP control or Lenti-shRNA-*Efnb2* into cervical ventral horn (**a**). Injections were made bilaterally into six sites throughout C3-C5 to target the PhMN pool (**b**). Thirty μm transverse tissue sections of the cervical spinal cord show robust expression of the GFP reporter bilaterally within the ventral horn (**c**); scale bar: 500 μm. Cell lineage of viral transduction was assessed using three markers: neuronal marker NeuN, oligodendrocyte lineage marker Olig2, and astrocyte marker GFAP. Spinal cord tissue collected from SOD1[G93A] injected with Lenti-GFP vector was sectioned at 30 μm and immunostained for NeuN (**d-f**), Olig2 (**g–i**) and GFAP (**j–l**); scale bar: 150 μm. Quantification of transduction lineage was assessed by counting the total numbers of GFP+ cells that were co-labeled with each lineage-specific marker and expressing this as a percentage of the total number of GFP+ cells (**m**). We assessed the amount of knockdown achieved by the Lenti-shRNA-*Efnb2* vector by immunostaining endstage SOD1[G93A] cervical spinal cord tissue with an anti-ephrinB2 antibody in both Lenti-GFP control (**o, q**) and Lenti-shRNA-*Efnb2* (**p, r**) tissue; scale bar: 100 μm. Knockdown was quantified by counting the total number of GFP+ cells expressing ephrinB2+ within the cervical ventral horn (**n**). Analyses in all panels: n=3 mice per condition; 1 female and 2 males per condition.

The online version of this article includes the following source data for figure 2:

**Source data 1.** File contains the raw data for *Figure 2*, panels M and N.

## Protection of MNs in the cervical spinal cord

Loss of MNs in spinal cord is a hallmark of ALS. To determine whether knockdown of ephrinB2 in astrocytes might impact MN survival selectively in the region of ephrinB2 knockdown, we quantified MN somata within the C3-5 spinal cord. Using cresyl violet staining of transverse cervical spinal cord sections, the number of neurons with a somal diameter greater than 20 μm and with an identifiable nucleolus was determined (MNs, *Figure 3a*; *Lepore et al., 2008b*). In C3, C4, and C5 following transduction of Lenti-shRNA-*Efnb2* (*Figure 3d*), there was a significantly greater number of MNs within the ventral horn compared to Lenti-GFP controls (*Figure 3c*) (Lenti-GFP: 266.4±19.46 MNs/μm$^2$, n=4 mice; Lenti-shRNA-*Efnb2*: 344.3±6.31 MNs/μm$^2$, n=4 mice; p=0.019, t-test) (*Figure 3b*). These data suggest that knockdown of ephrinB2 can increase survival of MNs in a mutant SOD1 model of ALS.

## Preservation of diaphragm function

Patients ultimately succumb to ALS because of respiratory compromise due significantly in part to loss of PhMNs that innervate diaphragm, the primary muscle of inspiration (*Mitsumoto et al., 1998*). To evaluate whether ephrinB2 knockdown in astrocytes focally within the PhMN pool impacts respiratory neural circuitry, we determined effects on both PhMN innervation of diaphragm using morphological assessment (*Nicaise et al., 2013*; *Nicaise et al., 2012a*; *Nicaise et al., 2012b*) and preservation of diaphragm function using in vivo electrophysiological measurements (*Lepore et al., 2008b*; *Nicaise et al., 2013*; *Lepore et al., 2011b*; *Lepore et al., 2010*). In anesthetized mice, we recorded compound muscle action potential (CMAP) amplitudes from each hemi-diaphragm following supramaximal stimulation of the ipsilateral phrenic nerve, an electrophysiological assay of functional diaphragm innervation by PhMNs. We performed these experiments in SOD1$^{G93A}$ mice at 117 days of age, a time point following the beginnings of forelimb motor dysfunction in the vast majority of animals but prior to endstage. Quantification of CMAP amplitude showed a 61% larger amplitude for the lenti-shRNA group (*Figure 3f*) compared to lenti-GFP (*Figure 3e–g*), demonstrating that ephrinB2 knockdown in cervical ventral horn resulted in significant preservation of functional diaphragm innervation (Lenti-GFP: 2.58±0.26 mV, n=4 mice; Lenti-shRNA: 4.20±0.32 mV, n=4 mice; t-test, p=0.0075). These data indicate that region-specific knockdown of ephrinB2 was able to generate functional rescue appropriate for the location targeted.

## Effects on disease onset, disease duration, and animal survival

We chose to perform anatomically-targeted shRNA delivery to only the ventral horn of the cervical (C3-C5) spinal cord in order to specifically target the critically-important phrenic nucleus and to use this motor circuit as a model system to examine the impact of knocking down astrocyte ephrinB2 expression on PhMN degeneration and diaphragm innervation. As expected, given that injections were delivered only to levels C3-5, ephrinB2 knockdown in astrocytes had no impact on overall disease phenotype, including limb motor function, disease onset and progression, and animal survival, as assessed by a battery of established measurements (*Lepore et al., 2008b*; *Lepore et al., 2011b*; *Lepore et al., 2010*; *Lepore et al., 2007*). EphrinB2 knockdown did not affect weight loss at any age tested (F $_{(1, 18)}$=0.17, p=0.69) (*Figure 4a*; n=8–10 mice per group). Additionally, overall disease onset as determined by the timing of weight loss onset was unaffected, with both the lenti-GFP and lenti-shRNA groups showing similar onset as determined by Kaplan-Meier analysis (Lenti-GFP: 124.4 days; Lenti-shRNA-*Efnb2* 123.5 days, chi square: 0.017, p=0.90, Gehan-Breslow-Wilcoxon test; n=8–9 mice per group) (*Figure 4b*). Furthermore, there were no differences between the two groups in either hindlimb (F $_{(1, 18)}$=0.48, p=0.50, ANOVA; n=9–10 mice per group) (*Figure 4c*) or forelimb (F $_{(1, 18)}$=0.95, p=0.34, ANOVA; n=9–10 mice per group) (*Figure 4e*) grip strength decline. We also used these grip strength measurements to calculate hindlimb and forelimb disease onsets. We calculated onset individually for each animal as the age with a 10% decline in grip strength compared to the maximum strength for those limbs in the same animal (*Lepore et al., 2011b*; *Lepore et al., 2007*). EphrinB2 knockdown had no effect on either hindlimb disease onset (Lenti-GFP: 90.4 days; Lenti-shRNA-*Efnb2* 103.0 days, chi square: 2.92, p=0.09, Gehan-Breslow-Wilcoxon test; n=9–10 mice per group) (*Figure 4d*) or forelimb disease onset (Lenti-GFP: 107.2 days; Lenti-shRNA-*Efnb2* 105.1 days, chi square: 0.13, p=0.72, Gehan-Breslow-Wilcoxon test; n=9–10 mice per group) (*Figure 4f*). Given that previous work showed that astrocytes contribute to disease progression in mutant SOD1 rodents post-disease onset (*Yamanaka et al., 2008*), we examined whether ephrinB2 knockdown in astrocytes extended disease

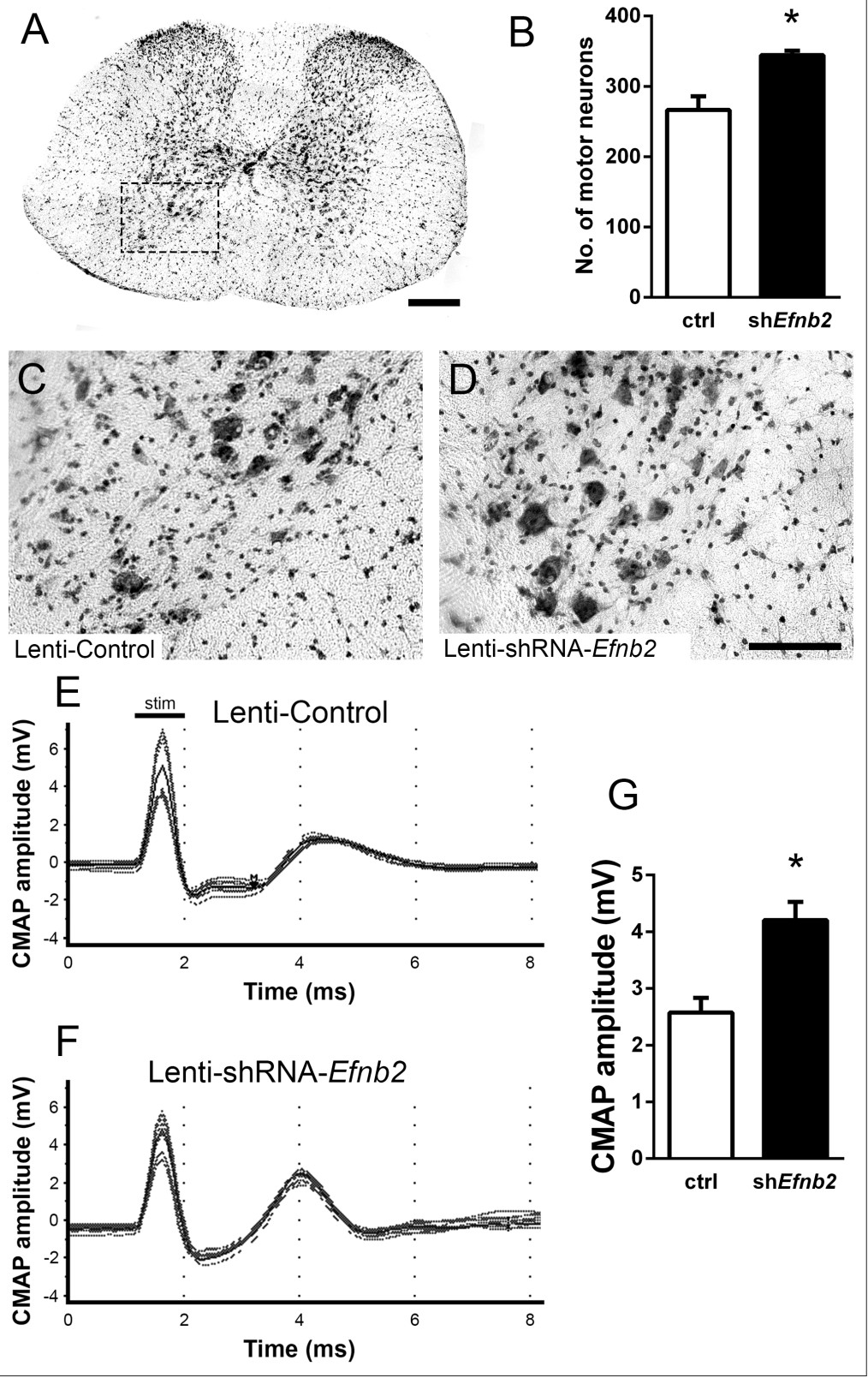

**Figure 3.** EphrinB2 knockdown protected cervical spinal cord motor neurons and preserved functional innervation of the diaphragm in SOD1$^{G93A}$ mice. SOD1$^{G93A}$ mice injected with Lenti-GFP control or Lenti-shRNA-*Efnb2* were cresyl violet stained and MN counts were performed at 117 days of age. Thirty μm transverse cervical spinal cord tissue sections were stained with cresyl violet (**a**); scale bar: 250 μm. The dotted box outlines the ventral horn and

*Figure 3 continued on next page*

*Figure 3 continued*

area of the image shown in (**c**). MN populations within the ventral horn were quantified (**b**). Representative images show a greater loss of MNs in Lenti-Control (**c**) compared to the Lenti-shRNA-*Efnb2* (**d**) group; scale bar: 100 μm. SOD1^G93A mice injected with Lenti-GFP control or Lenti-shRNA-*Efnb2* were assessed in vivo for PhMN-diaphragm innervation by electrophysiological analysis at 117 days of age. CMAP amplitudes were recorded from each hemi-diaphragm following ipsilateral phrenic nerve stimulation. Representative traces of Lenti-GFP (**e**) and Lenti-shRNA-*Efnb2* (**f**) recordings show a larger CMAP amplitude in the Lenti-shRNA-*Efnb2* group. Quantification of maximal CMAP amplitude shows significant preservation in the Lenti-shRNA-*Efnb2*-treated group compared to control (**g**). Analysis in panels A-D: n=4 mice per condition; 2 females and 2 males per condition. Analysis in panels E-G: n=4 mice per genotype and per time point; 2 females and 2 males per condition.

The online version of this article includes the following source data for figure 3:

**Source data 1.** File contains the raw data for *Figure 3*, panels B and G.

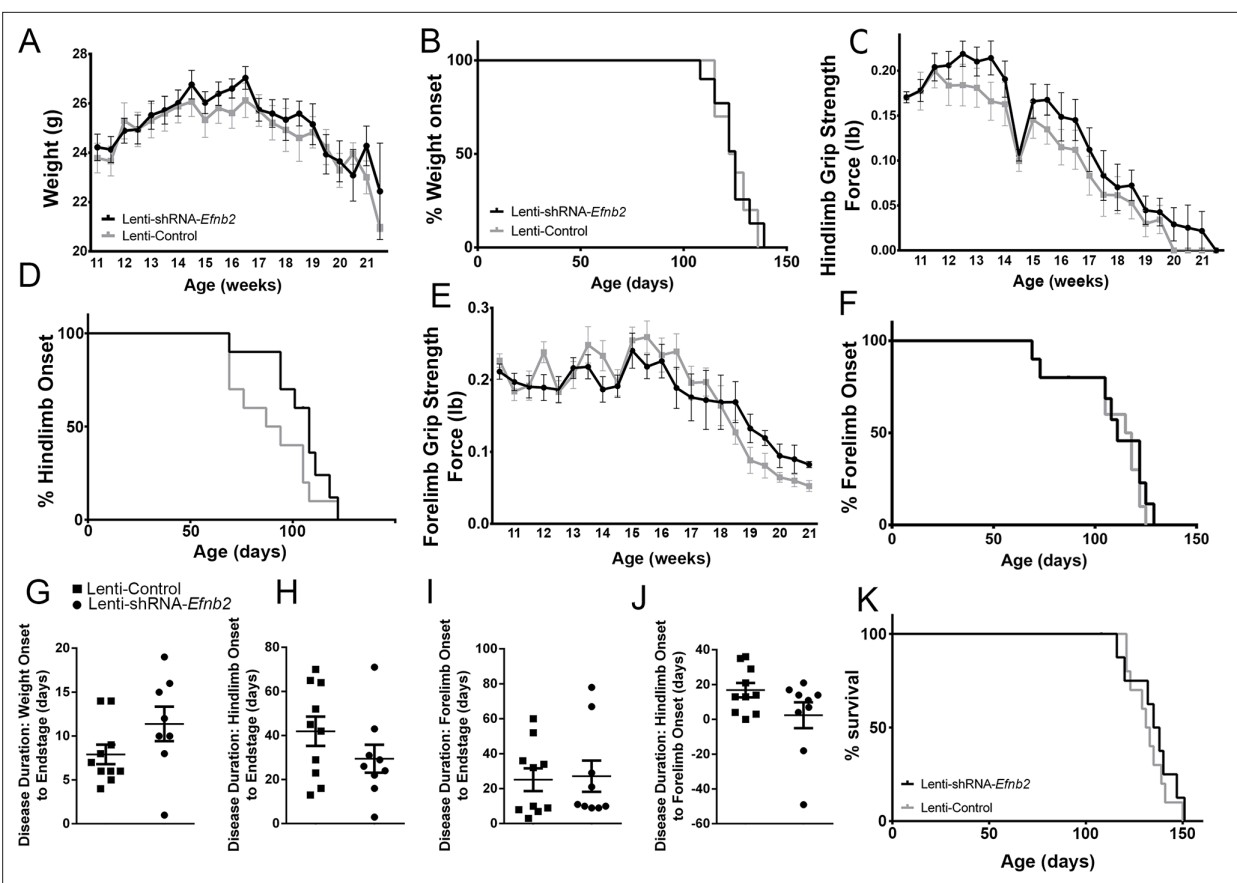

**Figure 4.** Knockdown of ephrinB2 in the cervical spinal cord ventral horn did not extend survival or delay onset of disease in SOD1^G93A mice. Biweekly weights of Lenti-shRNA-*Efnb2* and Lenti-Control mice were recorded until endpoint sacrifice (**a**), and disease onset was measured for each animal when there was a 10% drop in body weight (**b**). Biweekly individual hindlimb grip strengths were assessed for each animal (**c**), and disease onset was recorded when the animal had a 10% decline in hindlimb grip strength (**d**). Each animal was also tested for forelimb grip strength (**e**), and disease onset was recorded when the animal had a 10% decline in forelimb grip strength (**f**). Weights, forelimb grip strength and hindlimb grip strength were taken biweekly starting one week prior to injection of Lenti-shRNA-*Efnb2* or Lenti-GFP control, and all force measurements plotted were the average force (lb) of all animals combined in each group. Disease duration was determined by time from weight onset to endstage (**g**), hindlimb disease onset to endstage (**h**), and forelimb diseaseonset to endstage for each animal (**i**). Disease duration was also measured from the time of forelimb disease onset to time of hindlimb disease onset (**j**). Survival was measured as the day each animal reached endstage, which was determined by the righting reflex (**k**). Analyses in all panels: n=8–10 mice per genotype and per time point; 4–5 females and 4–5 males per condition.

The online version of this article includes the following source data for figure 4:

**Source data 1.** File contains the raw data for *Figure 4*, panels A-K.

duration. Compared to lenti-GFP control, lenti-shRNA had no effect on disease duration as measured by the time from: weight onset to endstage (Lenti-GFP: 7.90±1.110 days, n=10 mice; Lenti-shRNA-*Efnb2*: 11.38±1.963 days, p=0.13, unpaired t-test; n=8 mice) (*Figure 4g*); hindlimb disease onset to endstage (Lenti-GFP: 41.90±6.63 days, n=10 mice; Lenti-shRNA-*Efnb2*: 29.44±6.34 days, n=9 mice, p=0.19, unpaired t-test) (*Figure 4h*); forelimb disease onset to endstage (Lenti-GFP: 25.10±6.48 days, n=10 mice; Lenti-shRNA-*Efnb2*: 27.11±8.92 days, n=9 mice, p=0.86, unpaired t-test) (*Figure 4i*); or hindlimb disease onset to forelimb disease onset (Lenti-GFP: 16.80±4.14 days, n=10 mice; Lenti-shRNA-*Efnb2*: 2.33±7.44 days, n=9 mice, p=0.099, unpaired t-test) (*Figure 4j*). Lastly, given that we targeted the location of the critically-important pool of PhMNs with our virus injections, we determined whether ephrinB2 knockdown specifically within the cervical ventral horn extended animal survival, as determined by the righting reflex (*Lepore et al., 2011b*; *Lepore et al., 2007*). Compared to lenti-GFP control, lenti-shRNA had no effect on the age of disease endstage as determined by Kaplan-Meier analysis (Lenti-GFP: 132.3 days; Lenti-shRNA-*Efnb2* 129.5 days, chi square: 0.24, p=0.63, Gehan-Breslow-Wilcoxon test; n=8–10 mice per group) (*Figure 4k*).

## Preservation of PhMN innervation of the diaphragm

We next quantified morphological innervation changes at the diaphragm NMJ, as this synapse is critical for functional PhMN-diaphragm circuit connectivity. We labeled phrenic motor axons and their terminals for neurofilament (using SMI-312R antibody) and synaptic vesicle protein 2 (SV2), respectively, and we labeled nicotinic acetylcholine receptors with Alexa555-conjugated alpha-bungarotoxin (*Nicaise et al., 2012a*; *Lepore et al., 2010*). Using confocal imaging of individual NMJs, we quantified the percentage of intact (*Figure 5a*), partially-denervated (*Figure 5b*) and completely-denervated (*Figure 5c*) NMJs in the diaphragm (*Wright et al., 2007*; *Wright et al., 2009*; *Wright and Son, 2007*). Although we assessed NMJ morphology only in SOD1$^{G93A}$ mice in this work, our previous findings show that all diaphragm NMJs in non-diseased WT mice are completely intact (*Nicaise et al., 2012b*; *Lepore et al., 2010*; *Li et al., 2015b*; *Martin et al., 2015*). In the current study, we find extensive denervation at a large portion of NMJs across the diaphragm at 117 days of age in SOD1$^{G93A}$ mice. Compared to control-treated animals (*Figure 5d*), the lenti-shRNA group (*Figure 5e*) showed a significant increase in the percentage of fully-innervated NMJs (*Figure 5f*) and a significant decrease in percentage of completely-denervated junctions (*Figure 5g*), demonstrating that lenti-shRNA treatment preserved PhMN innervation of the diaphragm (innervated: Lenti-GFP: 27.0 ± 2.5% of total NMJs, n=4 mice; Lenti-shRNA: 53.2±8.5, n=4; t-test, p=0.04) (denervated: Lenti-GFP: 21.6 ± 1.6% of total NMJs, n=4 mice; Lenti-shRNA: 8.0±3.7, n=4 mice; t-test, p=0.03). We also found a trend toward a decrease in the percentage of partially-denervated NMJs in lenti-shRNA animals versus control (*Figure 5h*), though the difference was not significant (partially-denervated: Lenti-GFP: 42.1 ± 0.7% of total NMJs, n=4 mice; Lenti-shRNA: 29.6±6.2; n=4 mice, t-test, p=0.11). Our NMJ analyses suggest that preservation of diaphragm innervation by PhMNs with focally-delivered lenti-shRNA-*Efnb2* resulted in a maintenance of diaphragm function. The increased cervical MN survival in the lenti-shRNA-*Efnb2* group coincided with enhanced preservation of diaphragm NMJ innervation, suggesting that the ephrinB2 knockdown-mediated effects on NMJ innervation and CMAP amplitudes were due at least in part to protection of PhMNs centrally within the cervical spinal cord.

## EphrinB2 upregulation in human ALS spinal cord

We also performed immunoblotting analysis on postmortem samples from human ALS donors with an SOD1 mutation (n=3 donors) and matching non-diseased human samples (n=3 donors). In the lumbar enlargement, there was a large increase in ephrinB2 protein expression in the SOD1 mutation ALS samples compared to the non-diseased controls (*Figure 6*) (non-ALS lumbar: 3.8±1.1 a.u.; ALS lumbar: 17.8±18.2; ALS cortex: 2.9±0.25; p=0.523, ANOVA). There was some donor-to-donor variability; while all of the non-ALS control samples showed similarly lower levels of ephrinB2 protein expression in the lumbar spinal cord, dramatic ephrinB2 upregulation in the SOD1 mutation samples was observed with only two of the three ALS donors. The absence of ephrinB2 upregulation in the one ALS sample may be related to the anatomical progression of disease in this particular donor. To this point, we also performed GFAP immunoblotting on the same lumbar spinal cord samples and found signficantly higher GFAP protein levels in the two samples with increased ephrinB2 expression (*Figure 6*). As the level of GFAP expression is often used as an indicator of disease progression at

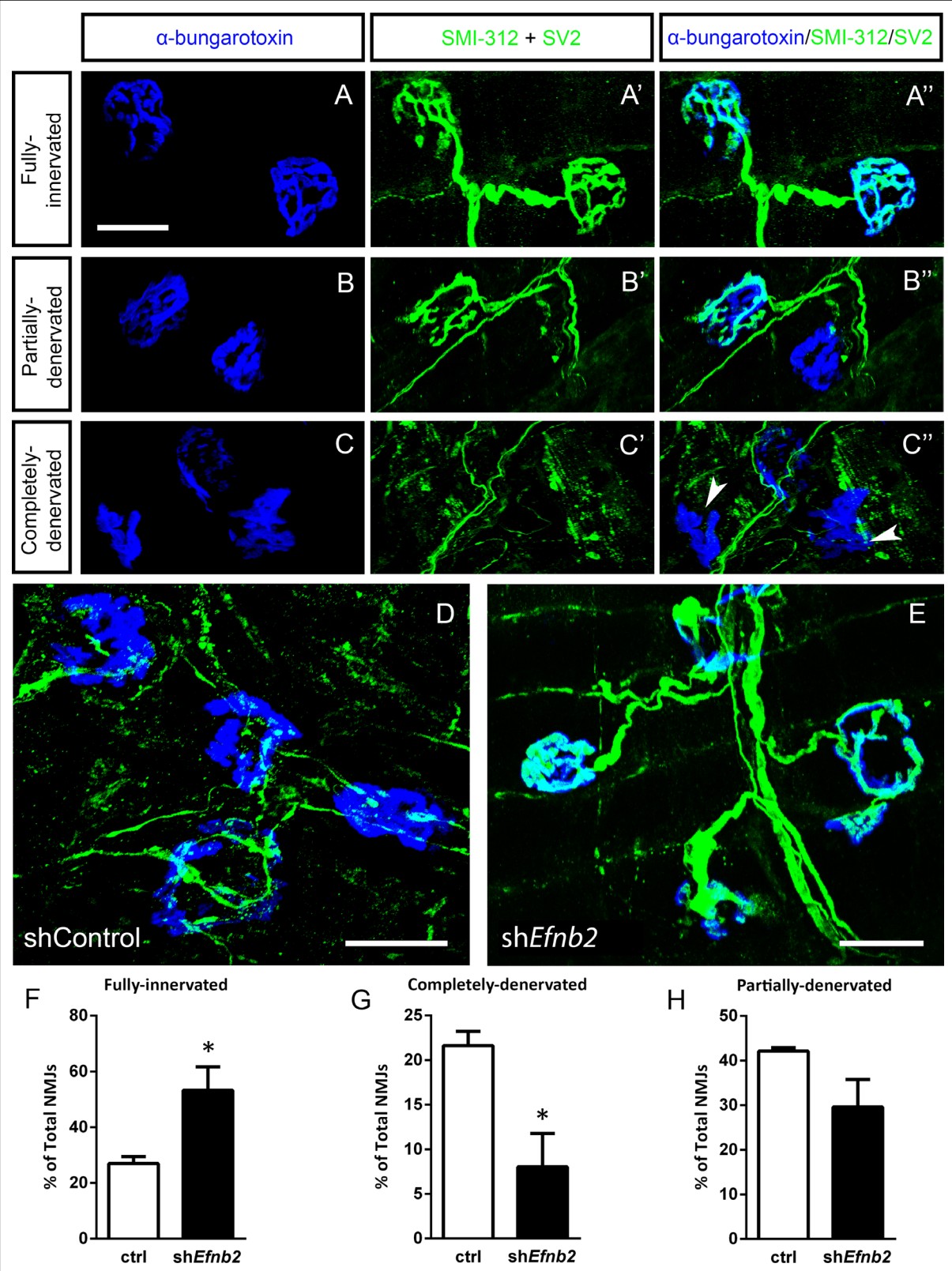

**Figure 5.** EphrinB2 knockdown preserved morphological innervation of the diaphragm in SOD1$^{G93A}$ mice. Diaphragm muscles were labeled with SMI-312 (green), SV2, (green) and alpha-Bungarotoxin (blue). Representative images of fully-innervated (**a**), partially-denervated (**b**) and completely-denervated ((**c**): arrowheads denote completely-denervated NMJs) NMJs are shown; all scale bars: 30 μm. Compared to SOD1$^{G93A}$ mice treated with Lenti-GFP (**d**), animals injected with Lenti-shRNA-*Efnb2* (**e**) showed greater preservation of PhMN innervation of the diaphragm NMJ. Quantification

*Figure 5 continued on next page*

*Figure 5 continued*

revealed a significant increase in the percentage of fully-innervated NMJs (**f**) and a decrease in the percentage of completely-denervated NMJs (**g**) in the Lenti-shRNA-*Efnb2* group compared to Lenti-GFP controls. The percentage of partially-denervated NMJs was not statistically different between the two groups (**h**). Analyses in all panels: n=4 mice per condition; 2 females and 2 males per condition.

The online version of this article includes the following source data for figure 5:

**Source data 1.** File contains the raw data for *Figure 5*, panels F-H.

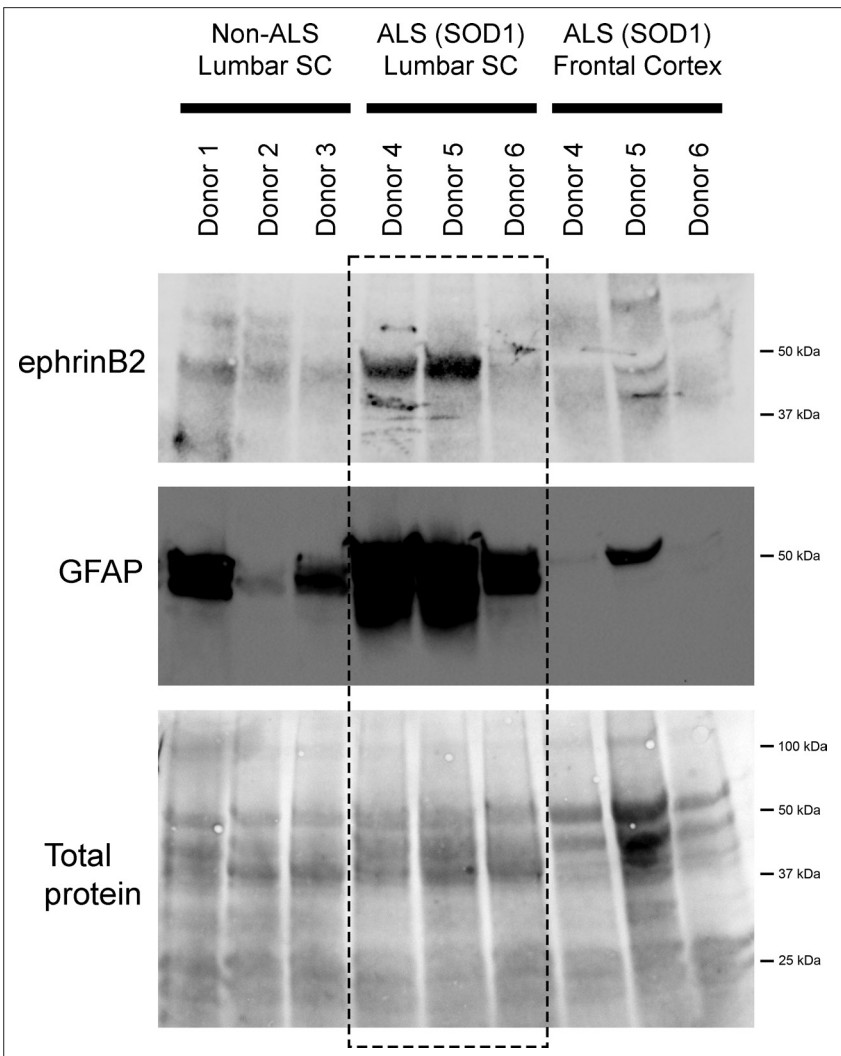

**Figure 6.** EphrinB2 upregulation in human spinal cord from ALS donors with an SOD1 mutation. Immunoblotting analysis on postmortem samples from human ALS donors with an SOD1 mutation and also non-diseased human samples. In the lumbar enlargement, there was a large increase in ephrinB2 protein expression in the SOD1 mutation ALS samples compared to the non-diseased controls (top blot). GFAP immunoblotting on the same lumbar spinal cord samples shows a robust increase in GFAP protein levels in the two samples with increased ephrinB2 expression (middle blot). There was not increased ephrinB2 expression in a disease unaffected region in these same three ALS donor samples, as ephrinB2 protein levels were not elevated in the frontal cortex (top blot). Immunoblot for total protein (bottom blot). Demographic information: Donor 1 – death at 67 years; male; non-ALS; Donor 2 – death at 70 years; male; non-ALS; Donor 3 – death at 70 years; female; non-ALS; Donor 4 – death at 42 years; female; SOD1-D102H mutation; absence of C9orf72 repeat expansion; Donor 5 – death at 55 years; male; SOD1-A4V mutation; absence of C9orf72 repeat expansion; Donor 6 – death at 58 years; male; SOD1-V87A mutation; absence of C9orf72 repeat expansion.

The online version of this article includes the following source data for figure 6:

**Source data 1.** File contains the raw data for *Figure 6*.

a particular anatomical region, this finding suggests that ephrinB2 upregulation may have occurred selectively at locations in the CNS where disease processes were already occurring by the time of death. Lastly, we did not observe increased ephrinB2 expression in a disease unaffected region in these same three ALS donor samples, as ephrinB2 protein levels were not elevated in frontal cortex (*Figure 6*).

## Discussion

We have shown that ephrinB2 is expressed predominantly by ventral horn astrocytes and that ephrinB2 up-regulation coincided with progression of MN loss and overall disease phenotype in the SOD1$^{G93A}$ mouse model of ALS. Furthermore, we found that reducing ephrinB2 expression in ventral horn astrocytes of the SOD1$^{G93A}$ mouse cervical spinal cord maintained diaphragmatic respiratory function by protecting cervical spinal cord MNs and preserving PhMN innervation of the diaphragm. Despite the significant impact that ephrinB2 knockdown had on functional diaphragm innervation, we did not observe effects on limb motor function, disease onset, phenotypic progression of disease post-onset, or overall animal survival. Measures such as disease duration and animal survival also depend on other MN populations such as those present in lumbar spinal cord, brainstem and motor cortex, while our targeted strategy only addresses cervical MN loss (and more specifically those MNs located only at C3-C5). That being said, PhMN preservation plays a critical role in both SOD1$^{G93A}$ models and human disease (*Lladó et al., 2006*), and we have previously shown that focal protection of cervical enlargement MNs using glial progenitor transplantation does extend overall disease phenotype in SOD1$^{G93A}$ rodents (*Lepore et al., 2008b*). Nevertheless, in future work aimed at addressing more translational considerations, we can extend this approach to delivery strategies such as intrathecal injection to target ephrinB2 throughout the spinal cord neuraxis. These data with focal ephrinB2 knockdown demonstrate the important role played by ephrinB2 in MN health and muscle function in mutant SOD1-associated ALS pathogenesis and suggest that ephrinB2 is a promising target for further investigation.

Other than in a relatively small number of studies (*Van Hoecke et al., 2012*; *Rué et al., 2019a*; *Ling et al., 2018*; *Rué et al., 2019b*; *Zhao et al., 2018*; *Tsuda et al., 2008*; *Tyzack et al., 2017*; *Zhao et al., 2017*), the role of Eph-ephrin signaling in ALS has not been extensively examined. Our findings suggest that astrocyte ephrinB2 may play a non-cell autonomous role in ALS, in particular in mutant SOD1-associated disease. A substantial body of work has demonstrated that astrocytes are involved in ALS pathogenesis (*Ilieva et al., 2009*; *Yamanaka and Komine, 2018*), both via loss of critically important functions such as extracellular glutamate uptake (*Rothstein et al., 1995*) and toxic gain-of-function such as altered transforming growth factor β (TGFβ) signaling (*Phatnani et al., 2013*) and increased production of reactive oxygen species (ROS) (*Cassina et al., 2008*). The dramatic increase of ephrinB2 expression observed in ventral horn astrocytes may represent an additional toxic property.

What is the mechanism by which astrocyte ephrinB2 contributes to MN pathology in ALS? EphA4 receptor expression and signaling capacity correlate with the degree of human ALS disease severity, and EphA4 significantly contributes to MN degeneration in several animal models of ALS (*Van Hoecke et al., 2012*). EphA4 receptor can be activated by ephrinA and ephrinB ligands including ephrinB2 (*Flanagan and Vanderhaeghen, 1998*), suggesting that astocyte ephrinB2 serves as a ligand mediating pathogenic actions in ALS. Consistent with this model, ephrinB2 is upregulated in the SOD1$^{G93A}$ mouse model and reduction of ephrinB2 expression increases MN survival near the site of ephrinB2 knockdown. Going forward, we could explore this potential interaction in vivo and in astrocyte-MN co-cultures using approaches such cell type-specific knockout of ephrinB2 and EphA4 expression, preventing ephrinB2-EphA4 binding, and EphA4 receptor kinase activity indicators.

Previous findings suggest that EphA-ephrinB signaling may contribute to ALS pathogenesis. Initial results showed that EphA4 plays a significant role in both ALS animal models and in the human ALS population (*Van Hoecke et al., 2012*). Subsequent work in mouse models (*Ling et al., 2018*; *Rué et al., 2019b*; *Zhao et al., 2018*) showed that genetic reduction of EphA4 in SOD1$^{G93A}$ mice or intra-cerebroventricular delivery of an antisense oligonucleotide directed against EphA4 in both SOD1$^{G93A}$ and PFN1$^{G118V}$ mouse models of ALS did not impact disease measures. In contrast, manipulations that target EphA4 signaling such as administration of an EphA4 agonist to mutant SOD1 mouse increased both disease duration and animal survival (*Wu et al., 2017*). In addition, delivery of soluble EphA4-Fc that blocks ligand binding to EphA4 results in partial preservation of motor function in SOD1$^{G93A}$

mice (*Zhao et al., 2018*). Importantly, these inhibitors act via blocking the Eph-ephrin interaction or disrupt bidirectional Eph-ephrin signaling. Thus, these data are consistent with a model where reverse or bidirectional EphA-ephrinB signaling via the interaction of EphA4 with ephrinB2 may be involved in ALS. Supporting this notion, genetic knockdown of ephrinA5 (an EphA4 ligand) in the SOD1^G93A mouse model accelerates disease progression and hastens animal death (*Rué et al., 2019a*), which may be explained by an enhancement of the EphA4-ephrinB2 interaction in the absence of ephrinA5. While other agents are also being developed to manipulate binding of ephrins with EphA4 for ALS therapeutics (*Qin et al., 2015*; *Schoonaert et al., 2017*), our results suggest that directly targeting ephrinB2 is a promising strategy to modulate both Eph-ephrin signaling and astrocyte-MN interactions in ALS. Unlike the effects of knocking down EphA4 expression in ALS animal models, we observe significant MN protection, maintenance of NMJ innervation and preservation of diaphragm muscle function following ephrinB2 reduction.

In addition to EphA4, ephrinB2 could signal via another Eph-family protein in ALS. Consistent with this model, ephrin-B upregulation in chronic pain models results in increased NMDA receptor function and pathological synaptic plasticity via interaction with EphBs (*Henderson and Dalva, 2018*). EphB's are centrally involved in regulating subcellular localization of ionotropic glutamate receptor subunits to excitatory synapses (*Hanamura et al., 2017*; *Nolt et al., 2011*), raising the intriguing possibility that enhanced ephrinB2-EphB2 signaling results in increased glutamate receptor activation in MNs and consequently may be contributing to excitotoxicity that plays a well-known role in ALS. Thus, ephrinB2 upregulation provides a number of potential avenues for aberrant circuit plasticity that could enhance neuronal damage and contribute to ALS pathogenesis.

In this study, we did not examine Eph receptor expression in the PhMN pool, but only focused on ephrinB2 in the surrounding astrocytes. Nevertheless, there are several pieces of evidence to suggest that PhMNs do express a variety of Eph receptors, which are possible candidates through which astrocyte ephrinB2 exerts its actions on PhMN health. EphrinB2 binds and activates EphBs, as well as EphAs such as EphA4. Importantly, previous studies have linked expression of EphA4 in MNs to the rate of ALS progression (*Van Hoecke et al., 2012*). Consistent with these studies, single-nucleus RNAseq on mouse cervical spinal cord shows that alpha MNs of the cervical spinal cord express various EphA and EphB receptors (http://spinalcordatlas.org/) (*Alkaslasi et al., 2021*; *Blum et al., 2021*). In addition, this dataset identifies a PhMN-specific marker (ErbB4); by specifically looking at the expression profile of only the ErbB4-expressing cervical alpha MNs, the data reveal that PhMNs express a number of EphA's and EphB's, including EphA4. To validate the expression specifically of EphA4, we find via IHC analysis that large ventral horn neurons are positive for phosphorylated EphA4 (a marker of activated EphA4) in C3-C5 spinal cord sections from the SOD1^G93A mice injected with shRNA-*Efnb2* vector or control vector (data not shown). These cervical spinal cord levels include MN pools in addition to just the PhMNs; therefore, this result by itself does not conclusively show that PhMNs at this location express EphA4, though they likely do since we find EphA4 expression in most large neuron cell bodies in C3-C5. Collectively, these RNAseq and IHC data show that PhMNs express a number of Eph receptors, which could be involved in direct cell-cell signaling with surrounding ephrinB2-expressing astrocytes.

EphrinB2 contribution to ALS is likely an astrocyte-mediated phenomenon given the pronounced upregulation that occurs almost entirely in ventral horn astrocytes. Nevertheless, our shRNA vector does not exclusively target only astrocytes. While the majority of transduced cells are astrocytes, we did not identify the lineage of a portion of the transduced cells, which could consist of cell types such as microglia (*Van Harten et al., 2021*), endothelial cells (*Mirian et al., 2022*) and others, some of which have been linked to ALS pathogenesis. It is therefore possible that the effects of ephrinB2 knockdown may not only be due to effects on astrocytes. However, we observed (1) dramatic ephrinB2 upregulation in the ventral horn of SOD1^G93A mice that appears to be astrocyte-specific, (2) transduction predominately of astrocytes with our vector, and (3) significant reduction of ephrinB2 expression almost entirely in astrocytes in shRNA-treated animals. These data suggest that the effect of *Efnb2*-shRNA treatment was primarily due to changes in astrocytes and that astrocyte expression is the likely mechanism underlying ephrinB2's action in mutant SOD1-associated ALS. In future work, could address this point by employing alternative vector-based strategies or cell type-specific knockout mouse models to selectively target astrocytes alone. In the present study, we used a viral vector with the U6 type-III RNA polymerase III promotor given its utility for persistently expressing high levels of

non-coding RNA transcripts such as shRNA (*Mäkinen et al., 2006*). Despite being a promotor that is not lineage-specific, we found that the majority of tranduced cells were GFAP-positive, although this transduction was still not completely limited to astrocytes. In our previous work, we instead used the *Gfa2* promotor to direct highly astrocyte-specific transduction in the adult rodent spinal cord (*Li et al., 2015b*; *Falnikar et al., 2016*; *Li et al., 2015a*; *Li et al., 2014*), which is an example of an approach moving forward to achieve astrocyte ephrinB2 knockdown in ALS animal models.

We previously showed that there is a modest increase in astrocyte number in ventral horn of the SOD1[G93A] mouse model at time points following phenotypic disease onset (*Lepore et al., 2008a*). It is therefore possible that the increased ephrinB2 expression observed across the ventral horn in SOD1[G93A] animals in the present study was due to this increased astrocyte number. However, this is unlikely to be the case, as astrocytes (and all other spinal cord cell types) in WT mice and in SOD1[G93A] mice prior to disease onset express very low levels of ephrinB2. Throughout disease course in these SOD1[G93A] mice, ephrinB2 level in individual astrocytes dramatically increases (including across most or all astrocytes), suggesting that the total increase in ephrinB2 expression across the ventral horn was not due to just this small increase in astrocyte numbers but was instead due to the dramatically elevated ephrinB2 expression observed across the astrocyte population.

In this study, we have shown that ephrinB2 protein expression is significanty increased in the lumbar spinal cord of postmortem samples from human ALS donors with an SOD1 mutation compared to non-diseased human samples, suggesting that this disease mechanism is also relevant to the human condition and is not restricted to only the mutant SOD1 mouse model. ALS is heterogeneous both with respect to its genetic basis and its clinical disease course (e.g. age and site of onset; severity/progression; *Mitsumoto et al., 1998*). The majority of patients have sporadic disease that is not linked to a known heritable genetic cause, while the remaining cases are linked to a known familial genetic mutation. Furthermore, these familial cases are associated with mutations in a number of different genes. In addition, patients (even with mutations in the same gene) show variability in their clinical disease manifestation such as the rate of disease progression depending on, for example, the specific SOD1 mutation (*Andersen et al., 1997*; *Radunovíc and Leigh, 1996*). However, the mechanisms underlying this heterogeneity in human disease progression are not understood. An important consideration is whether ephrinB2's function is specific to mutant SOD1-mediated disease or extends to more subtypes of ALS, including other disease-associated genes and sporadic ALS. To address whether ephrinB2 is a general modifier of ALS, future studies should focus on postmortem tissue samples and pluripotent stem cell-derived astrocytes and MNs derived from patients with various subtypes of the disease, as well as on animal models involving other ALS-associated genes. Previous work in ALS8-linked ALS (a form of familial ALS associated with the VAMP-associated protein B gene) suggests the possible relevance of Eph/ephrin biology to ALS pathogenesis (*Tsuda et al., 2008*). In addition, EphA4 knockdown can protect against the axonal damage response elicited by expression of ALS-linked mutant TDP-43 (*Van Hoecke et al., 2012*). These data suggest that altered Eph-ephrin signaling may not be limited to only mutant SOD1-assocated ALS, although more extensive investigation is necessary to support this idea.

Respiratory function involves the contribution of a number of other muscle groups besides diaphragm, and these muscles are innervated by various lower MN pools located across a relatively-large expanse of the CNS neuraxis. While CMAP recording is a powerful assay of functional innervation of diaphragm muscle by phrenic motor axons, it does not directly measure overall respiratory function. There are assays to test outcomes such as ventilatory behavior and gas exchange (e.g. whole-body plethysmography, blood gas measurements, etc.; *Nicaise et al., 2013*). As we focally targeted our ephrinB2 knockdown to only a small area (the phrenic nucleus), we would not expect an effect on these other functional assays, which is why we restricted our functional testing to CMAP recording to specifically study the effects of ephrinB2 knockdown on the PhMN pool.

Interestingly, we observed relatively robust effects of focal ephrinB2 knockdown in the cervical enlargement on functional diaphragm innervation, but did not similarly find effects on forelimb motor function using the forelimb grip strength assay, despite forelimb-innervating MN pools also residing in the cervical spinal cord. However, this grip strength motor assay is impacted primarily by distal forelimb muscle groups controlled by MN pools located at more caudal locations of the spinal cord (i.e. low cervical and high thoracic), likely explaining the lack of effect on grip strength. The localized – yet

robust – effects of ephrinB2 knockdown are consistent with the model that ephrinB2 is a target worth further exploration and validation.

In summary, we found astrocyte-specific upregulation of ephrinB2 expression in the ALS spinal cord, and we demonstrated that knocking down ephrinB2 in the ventral horn in an anatomically-targeted manner significantly preserved diaphragmatic respiratory neural circuitry in SOD1$^{G93A}$ mice. Importantly, ephrinB2 knockdown exerted significant protective effects on the centrally important population of respiratory PhMNs, which translated to maintenance of diaphragm function in vivo. We also report significantly increased ephrinB2 expression in the disease affected spinal cord of mutant SOD1 human ALS samples. In conclusion, our findings suggest that astrocyte ephrinB2 upregulation is both a signaling mechanism underlying astrocyte pathogenicity in mutant SOD1-associated ALS and a promising therapeutic target.

# Materials and methods

**Key resources table**

| Reagent type (species) or resource | Designation | Source or reference | Identifiers | Additional information |
|---|---|---|---|---|
| Genetic reagent (*Mus musculus*) | Transgenic SOD1$^{G93A}$ mice: C57BL/6 J congenic line B6.Cg-Tg(SOD1*G93A)1Gur/J | The Jackson Laboratory | RRID: IMSR_JAX:004435 | Both female and male mice used |
| Genetic reagent (*Mus musculus*) | Transgenic SOD1$^{G93A}$ mice: B6SJL-Tg(SOD1*G93A)1Gur/J | The Jackson Laboratory | RRID:IMSR_JAX:002726 | Both female and male mice used |
| Biological sample (*Homo sapiens*) | Spinal cord and cortex samples from ALS donors | Biorepository of the Jefferson Weinberg ALS Center (1 of 3 ALS samples) Project ALS (2 of 3 ALS samples) | | All three ALS donors had an SOD1 mutation (donor 1: D102H mutation; donor 2: A4V; donor 2: V87A), and all three donors did not have a C9orf72 repeat expansion. These three donors succumbed to ALS at 42 (female), 55 (male), or 58 (male) years of age. |
| Biological sample (*Homo sapiens*) | Spinal cord and cortex from non-ALS donors | NIH NeuroBioBank | | Age of death for these three non-ALS donors was 67, 70, and 70 years. |
| Antibody | Mouse monoclonal Anti-SV2 (Used for IHC) | Developmental Studies Hybridoma Bank, Iowa City, IA | RRID: AB_2315387 | 1:10 |
| Antibody | Mouse monoclonal Anti-SMI312 (Used for IHC) | Covance, Greenfield, IN | RRID: AB_2314906 | 1:1000 |
| Antibody | Mouse monoclonal Anti-NeuN (Used for IHC) | EMD-Millipore, Temecula, CA | RRID: AB_2298772 | 1:200 |
| Antibody | Rabbit polyclonal Anti-GFAP (Used for IHC) | Dako, Carpinteria, CA | RRID: AB_10013482 | 1:400 |
| Antibody | Mouse monoclonal Anti-GFAP (Used for Westerns) | BD Bioscience, Franklin Lakes, NJ | Cat. #610566 | 1:2000 |
| Antibody | Rabbit polyclonal Anti-Olig2 (Used for IHC) | EMD-Millipore, Temecula, CA | RRID: AB_2299035 | 1:200 |
| Antibody | Goat polyclonal Anti-ephrinB2 (Used for IHC) | R&D Systems, Minneapolis, MN | RRID: AB_2261967 | 1:50 |
| Antibody | Rabbit polyclonal Anti-ephrinB2 antibody (Used for Westerns) | Abcam, Cambridge, MA | RRID: AB_11156896 | 1:500 |

*Continued on next page*

*Continued*

| Reagent type (species) or resource | Designation | Source or reference | Identifiers | Additional information |
|---|---|---|---|---|
| Antibody | Goat polyclonal Anti-EphA4 (Used for IHC) | R&D Systems, Minneapolis, MN | RRID: AB_2099371 | 1:100 |
| Antibody | Rabbit polyclonal Anti-GFP (Used for IHC) | Aves Labs, Davis, CA | RRID: AB_10000240 | 1:500 |
| Recombinant DNA reagent | Lenti-shRNA-*Efnb2*; VSVG.HIV.SIN.cPPT.U6. SbRmEphrinB2.4.CMV.EGFP | This study | | Vector generated by the Dalva lab |
| Recombinant DNA reagent | Lenti-Control; VSVG.HIV.SIN.cPPT. U6.Empty.CMV.EGFP | This study | | Vector generated by the Dalva lab |
| Software | Scope 3.5.6 software | ADInstruments, Colorado Springs, CO | RRID: SCR_001620 | |
| Software | ImageJ/Fiji software | | RRID: SCR_003070 | |
| Software | Bio-Rad Image Lab software | Bio-Rad, Hercules, CA | RRID:SCR_014210 | |
| Software | Graphpad Prism 6 | Graphpad Software Inc, LaJolla, CA | RRID: SCR_002798 | |
| Other (Microscope) | Olympus FV1000 confocal microscope | Olympus, Center Valley, PA | RRID: SCR_014215 | 'NMJ analysis' subsection of the Materials and methods section |
| Other (Microscope) | Zeiss Axio M2 Imager confocal microscope | Zeiss, Hebron, KY | RRID:SCR_020922 | 'Motor neuron counts' subsection of the Materials and methods section |

## Animal model

Female and male transgenic SOD1$^{G93A}$ mice (C57BL/6 J congenic line: B6.Cg-Tg(SOD1*G93A)1Gur/J and B6SJL-Tg(SOD1*G93A)1Gur/J) were used in all experiments. All procedures were carried out in compliance with the National Institutes of Health (NIH) Guide for the Care and Use of Laboratory Animals and the ARRIVE (*Animal Research: Reporting of* In Vivo *Experiments*) guidelines. Experimental procedures were approved by Thomas Jefferson University Institutional Animal Care and Use Committee (IACUC) (approved IACUC protocol #01230). All animals were housed in a temperature-, humidity-, and light-controlled animal facility and were provided with food and water ad libitum.

## Endstage care

Due to the progression of muscle paralysis, animals were given access to softened food and were checked daily for overall health once the animals reached phenotypic onset of disease. We determined the onset for each animal by assessing total weight, hindpaw grip strength and forepaw grip strength (described below) (*Lepore et al., 2008b*; *Li et al., 2015b*). Animals were considered to have reached onset when there was a 10% loss in total body weight or a 10% loss in either forelimb or hindlimb grip strength. To determine the endstage for each animal, we used the 'righting reflex' method. We placed animals on their left and right sides; if a mouse could not right itself after 30 s on both sides, it was euthanized with an overdose of ketamine/xylazine.

## Viral vectors

Vectors used were VSVG.HIV.SIN.cPPT.U6.SbRmEphrinB2.4.CMV.EGFP and VSVG.HIV.SIN.cPPT.U6.Empty.CMV.EGFP. Lenti-shRNA-*Efnb2* or Lenti-Control constructs were driven by the U6 promoter, and EGFP expression was driven by the cytomegalovirus (CMV) promoter *McClelland et al., 2009*. shRNA sequence: *Efnb2* shRNA: 5-GCAGACAGATGCACAATTA-3. Forward and reverse oligonucleotides were synthesized (Integrated DNA Technologies) and generated a dsDNA insert consisting of forward and reverse complement RNAi sequences separated by a hairpin region and flanked by restriction site overhangs. We used 1.9x10^10 for intraspinal injections (described below).

## Intraspinal injection

For the intraspinal injections (*Goulão et al., 2019*; *Lepore, 2011a*; *Li et al., 2015c*), mice were first anesthetized with 1% isoflurane in oxygen, and the dorsal surface of the skin was shaved and cleaned with 70% ethanol. A half-inch incision was made on the dorsal skin starting at the base of the skull, and the underlying muscle layers were separated with a sterile surgical blade along the midline between the spinous processes of C2 and T1 to expose the cervical laminae. Paravertebral muscles overlying C3-C5 were removed using rongeurs, followed by bilateral laminectomies of the vertebrae over the C3-C5 spinal cord. A 33-gauge (G) needle on a Hamilton microsyringe (Hamilton, Reno, Nevada) was lowered 0.8 mm ventral from the dorsal surface just medial to the entry of the dorsal rootlets at C3, C4 and C5. After inserting the needle into the ventral horn, we waited three minutes before injecting the viral constructs. Two µL of Lenti-shRNA-*Efnb2* or Lenti-Control virus were delivered to the spinal cord over 5 min, controlled by an UltraMicroPump and Micro4 Microsyringe Pump Controller (World Precision Instruments, Sarasota, Florida). After injection, the needle was left in place for 3 min before being slowly removed. Following intraspinal injection, dorsal muscle layers were sutured with 4–0 silk sutures (Covidien, Minneapolis, Minnesota) and the skin was closed with surgical staples (Braintree Scientific, Braintree, Massachusetts). The surface of the skin was treated with a topical iodine solution. Immediately following the procedure, mice were given 1 mL of Lactated Ringer's solution (Hospira, San Jose, California) and cefazolin (6 mg) (Hospira, San Jose, California) via subcutaneous injections. Mice were placed in a clean cage on a surgical heating pad set to 37 ° C (Gaymar, Orchard Park, New York). At 12 and 24 hr after surgery, each animal was given an additional dose of buprenorphine hydrochloride (0.05 mg/kg) and monitored for pain/distress. Mice were 60 days old at the time of virus injection.

## Weight and grip-strength test

Weights were measured for each animal biweekly prior to forelimb and hindlimb testing. Forelimb and hindlimb grip strengths were determined using a 'Grip Strength Meter' (DFIS-2 Series Digital Force Gauge; Columbus Instruments, OH) (*Lepore et al., 2011b*; *Li et al., 2015c*). Grip strength was measured by allowing the animals to tightly grasp a force gauge bar using both forepaws or both hindpaws, and then pulling the mice away from the gauge until both limbs released the bar. The force measurements were recorded in three trials, and the averages were used in analyses. Grip strengths were recorded biweekly starting one week prior to initial injection.

## Compound muscle action potential recordings

At 117 days of age, mice were anesthetized with isoflurane (Piramal Healthcare, Bethlehem, Pennsylvania) at a concentration of 1.0–1.5% in oxygen. Animals were placed supine, and the abdomen was shaved and cleaned with 70% ethanol. Phrenic nerve conduction studies were performed with stimulation of the phrenic nerve via needle electrodes trans-cutaneously inserted into the neck region in proximity to the passage of the phrenic nerve (*Cheng et al., 2021*; *Ghosh et al., 2018*). A reference electrode was placed on the shaved surface of the right costal region. Phrenic nerve was stimulated with a single burst at 6 mV (amplitude) for a 0.5 ms duration. Each animal was stimulated between 10 and 20 times to ensure reproducibility, and recordings were averaged for analysis. ADI Power-lab8/30stimulator and BioAMPamplifier (ADInstruments, Colorado Springs, CO) were used for both stimulation and recording, and Scope 3.5.6 software (ADInstruments, Colorado Springs, CO; RRID: SCR_001620) was used for subsequent data analysis. Following recordings, animals were immediately euthanized, and tissue was collected (as described below).

## Diaphragm dissection

Animals were euthanized by an intraperitoneal injection of ketamine/xylazine diluted in sterile saline and then placed in a supine position. A laparotomy was performed to expose the inferior surface of the diaphragm. The diaphragm was then excised using spring scissors (Fine Science Tools, Foster City, California), stretched flat and pinned down on silicon-coated 10 cm dishes, and washed with PBS (Gibco, Pittsburgh, Pennsylvania). Diaphragms were then fixed for 20 min in 4% paraformaldehyde (Electron Microscopy Sciences, Hatfield, Pennsylvania). After washing in PBS, superficial fascia was carefully removed from the surface of the diaphragm with Dumont #5 Forceps (Fine Science Tools, Foster City, California). Diaphragms were then stained for NMJ markers (described below).

## Diaphragm whole-mount histology

Fresh diaphragm muscle was dissected from each animal for whole-mount immunohistochemistry, as described above (*Cheng et al., 2021*; *Ghosh et al., 2019*). Diaphragms were rinsed in PBS and then incubated in 0.1 M glycine for 30 min. Following glycine incubation, α-bungarotoxin conjugated to Alexa Fluor 555 at 1:200 (Life Technologies, Waltham, Massachusetts) was used to label post-synaptic nicotinic acetylcholine receptors. Ice-cold methanol was then added to the diaphragms for 5 min, and then diaphragms were blocked for 1 hr at room temperature in a solution of 2% bovine serum albumin and 0.2% Triton X-100 diluted in PBS (this solution was used for both primary and secondary antibody dilutions). Primary antibodies were added overnight at 4 ° C: pre-synaptic vesicle marker anti-SV2 at 1:10 (Developmental Studies Hybridoma Bank, Iowa City, Iowa; RRID: AB_2315387); neurofilament marker anti-SMI-312 at 1:1000 (Covance, Greenfield, Indiana; RRID: AB_2314906). The diaphragms were then washed and secondary antibody solution was added for 1 hr at room temperature: FITC anti-mouse IgG secondary (Jackson ImmunoResearch Laboratories, West Grove, PA; 1:100). Diaphragms were mounted with Vectashield mounting medium (Vector Laboratories, Burlingame, California), coverslips were added, and slides were stored at –20 °C.

## Neuromuscular junction (NMJ) analysis

At 117 days of age, labeled muscles were analyzed for the percentage of NMJs that were intact, partially-denervated or completely denervated (*Lepore et al., 2010*; *Wright et al., 2009*). Whole-mounted diaphragms were imaged on a FV1000 confocal microscope (Olympus, Center Valley, Pennsylvania; RRID: SCR_014215). We conducted NMJ analysis on the right hemi-diaphragm.

## Spinal cord and brain dissection

Animals were euthanized by an intraperitoneal injection of ketamine/xylazine diluted in sterile saline (as described above). Following diaphragm removal (described below), the animal was exsanguinated by cutting the right atrium and transcardially perfused with 0.9% saline solution (Thermo Fisher Scientific, Pittsburgh, Pennsylvania) then 4% paraformaldehyde (Electron Microscopy Sciences, Hatfield, Pennsylvania) to fix the tissue. Following perfusion, the spinal cord and brain were excised with rongeurs (Fine Science Tools, Foster City, California) and kept in a 4% paraformaldehyde solution overnight at 4 ° C, washed with 0.1 M Phosphate Buffer (Sodium Phosphate Dibasic Heptahydrate [Sigma-Aldrich, St. Louis, Missouri] and Sodium Monobasic Monohydrate [Sigma-Aldrich]), and placed in 30% sucrose (Sigma-Aldrich). A second group of animals was not perfused with 4% paraformaldehyde, and brain and spinal cord tissue were collected unfixed. Both fixed and unfixed samples were placed into an embedding mold (Polysciences Inc, Warrington, Pennsylvania) and covered with tissue freezing medium (General Data, Cincinnati, Ohio). Samples were then flash frozen in 2-methylbutane (Thermo Fisher Scientific, Pittsburgh, Pennsylvania) chilled in dry ice. Tissue was sectioned at 30 μm on a cryostat (Thermo Fisher Scientific, Philadelphia, Pennsylvania), placed on glass microscope slides (Thermo Fisher Scientific, Pittsburgh, Pennsylvania), and dried overnight at room temperature before freezing the samples at –20 ° C for long-term storage.

## Spinal cord histology/cresyl violet staining

Spinal cord tissue section slides were dried at room temperature for 2 hr. Following drying, slides were rehydrated in 3-min baths of xylene, 100% ethanol, 95% ethanol, 70% ethanol and dH$_2$O. To stain the tissue, slides were placed in an Eriochrome solution (0.16% Eriochrome Cyanine, 0.4% Sulfuric Acid, 0.4% Ferric Chloride in dH$_2$O) for 14 min, washed with tap water, placed in a developing solution (0.3% ammonium hydroxide in dH$_2$O) for 5 min, washed with dH$_2$O, and then placed into a cresyl violet solution (0.4% cresyl violet, 6% 1 M sodium acetate, 34% 1 M acetic acid) for 18 min. After staining, slides were dehydrated by being placed in baths of dH$_2$O, 70% ethanol, 95% ethanol, 100% ethanol and xylene. Slides were mounted with poly-mount xylene (Polysciences, Warrington, Pennsylvania), and cover slips were added. Slides were then kept at room temperature for storage and analysis.

## Immunohistochemistry

Prior to immunostaining, tissue sections were dried for 1 hr at room temperature. Antigen retrieval was performed using R&D Systems Protocol (R&D Systems, Minneapolis, Minnesota). Immediately after antigen retrieval, a hydrophobic pen (Newcomer Supply, Middleton, Wisconsin) was used to

surround the tissue sections. Slides were blocked/permeabilized for 1 hr at room temperature with a solution of 5% Normal Horse Serum (Vector Laboratories, Burlingame, California), 0.2% Triton X-100 (Amresco, Solon, Ohio), diluted in PBS (primary and secondary antibodies were diluted in this solution as well). Slides were then treated with primary antibody overnight at 4 ° C with the following antibodies: neuronal marker anti-NeuN at 1:200 (EMD-Millipore, Temecula, California; AB_2298772); astrocyte marker anti-GFAP at 1:400 (Dako, Carpinteria, California; RRID: AB_10013482); oligodendrocyte lineage marker anti-Olig-2 at 1:200 (EMD-Millipore; RRID: AB_2299035); anti-ephrinB2 at 1:50 (R&D Systems, RRID: AB_2261967); anti-ephA4 at 1:100 (R&D Systems RRID: AB_2099371); and anti-GFP at 1:500 (Aves Labs, Davis, California, RRID: AB_10000240). On the following morning, samples were washed 3 x in PBS, and secondary antibody solutions were added for 1 hr at room temperature: donkey anti-rabbit IgG H&L (Alexa Fluor 647) at 1:200 (Abcam, Cambridge, Massachusetts); donkey anti-mouse IgG H&L (Alexa Fluor 488) at 1:200 (Abcam, Cambridge, Massachusetts); Rhodamine (TRITC) AffiniPure donkey anti-goat IgG (H+L) at 1:200 (Jackson ImmunoResearch, West Grove, Pennsylvania). Following secondary antibody treatment, samples were washed in PBS and 2 drops of FluorSave reagent (Calbiochem, San Diego, California) were added to tissue sections, then slides were coverslipped (Thermo Fisher Scientific, Pittsburgh, Pennsylvania). Slides were stored at 4 ° C.

## Viral vector transduction quantification

SOD1^G93A mouse cervical spinal cord tissue at disease endstage was immunostained with anti-GFP and either anti-GFAP, anti-NeuN or anti-Olig2 (described above). We quantified the percentage of double-labeled GFP+/GFAP+, GFP+/NeuN+ or GFP+/Olig2+ cells versus the total number of GFP+ cells in the ventral horn. The cell lineage of lenti-viral transduction was plotted as a percentage of the total GFP+ cells.

## Motor neuron counts

At 117 days of age, 30 μm mouse cervical spinal cord tissue sections were stained with cresyl violet (as described above) to determine the total number of MNs. Images were acquired using a 10 x objective on a Zeiss Axio M2 Imager (Carl Zeiss Inc, Thornwood, New York), and analyzed with ImageJ/Fiji software (RRID: SCR_003070). The area (converted into pixels) of each ventral horn was outlined separately starting from the central canal and tracing laterally and ventrally to encompass the right and left ventral horns for each spinal cord section. Within the area of each ventral horn, neurons were traced and somal area was assessed. We considered an MN as any neuron within the ventral horn greater than 20 μm in somal diameter and with an identifiable nucleolus (*Li et al., 2015b*). We then assessed total number of MNs per area of the ventral horn for both the Lenti-shRNA-*Efnb2* group and the Lenti-control group.

## EphrinB2 quantification

EphrinB2 levels in ventral horn of the cervical spinal cord of endstage SOD1^G93A mice intraspinally injected with Lenti-shRNA-*Efnb2* or Lenti-Control were evaluated. In addition, this same analysis was performed on uninjected SOD1^G93A mice at 60 days of age, 120 days of age, and at disease endstage, as well as on uninjected WT mice at 140 days of age. Thirty μm cervical spinal cord sections were immunostained with anti-GFP and anti-ephrinB2 antibodies. ShRNA-induced knockdown was assessed by quantifying the number of ephinB2+/GFP+ cells for both Lenti-GFP control and Lenti-shRNA-*Efnb2* groups. Four animals were used for each group, with the number of ephrinB2/GFP+ cells per animal averaged over three slides (eight tissue sections each).

## Human postmortem tissue

For analysis of human postmortem tissue, we examined three non-ALS and three ALS donors. Non-diseased samples were obtained from the NIH NeuroBioBank. Age of death for these three non-ALS donors was 67, 70, and 70 years. For the ALS samples, all three donors had an SOD1 mutation (donor 1: D102H mutation; donor 2: A4V; donor 2: V87A) and all did not have a C9orf72 repeat expansion. Two of these SOD1 ALS samples were obtained from Project ALS, and the third sample was obtained from the biorepository of the Jefferson Weinberg ALS Center. These three donors succumbed to ALS at 42 (female), 55 (male), or 58 (male) years of age.

## Immunoblotting of postmortem tissue

A total of 100 mg of fresh-frozen human autopsy sample (lumbar spinal cord or frontal cortex) were homogenized in 1% SDS using a Dounce homogenizer. Homogenate was centrifuged at 3000 rpm for 20 min at 4 °C to remove debris. Clear supernatant was then used to estimate total protein content using the bicinchoninic acid (BCA) assay (Pierce BCA kit #23225; Thermo Fischer Scientific, Waltham, Massachusetts). Thirty μg of protein were loaded onto 10% stain-free gel (#4568034; Bio-Rad, Hercules, California). After the run, gels were activated using UV light to crosslink protein and transferred to 0.22 μm nitrocellulose membrane. After transfer, membrane was exposed to chemiluminescence light to image total protein. Membrane was then blocked using 5% fat-free milk in tris-buffered aaline with tween (TBST) for one hour at room temperature. Anti-ephrinB2 antibody (Cat# ab131536, RRID: AB_11156896; Abcam, Cambridge, Massachusetts) at 1:500 dilution in 5% bovine serum albumin in TBST was incubated overnight, followed by three washes with TBST on the shaker for 15 min each. Anti-rabbit horseradish peroxidase (HRP) secondary (#NA9340V, Sigma-Aldrich, St. Louis, Missouri) at 1:5000 dilution was prepared in 5% fat-free milk and added to membrane for one hour at room temperature with shaking. Membranes were washed 3 x for 15 min on a shaker with TBST. Chemiluminescence signal was imaged using super signal west Atto (#38554; Bio-Rad, Hercules, California). The same membrane was used to probe for GFAP using anti-GFAP antibody (#610566; BD Bioscience, Franklin Lakes, New Jersey) at 1:2000 dilution overnight. Membrane was washed 3 x the next day with TBST and incubated with anti-mouse HRP (#NXA931V; Sigma-Aldrich, St. Louis, Missouri) at 1:5000 dilution for 1 hr at RT and washed, and then chemiluminescence was imaged as described above. Quantification for ephrinB2 was performed by normalizing to total protein using Bio-rad Image Lab software (RRID:SCR_014210).

## Reagents

We authenticated relevant experimental regents to ensure that they performed similarly across experiments and to validate the resulting data. Whenever we used a new batch of the vector, we verified that the virus performed equivalently from batch-to-batch by confirming in every animal that the vector transduced predominantly GFAP-positive astrocytes and induced similar expression of the GFP reporter for each batch. For Alexa-conjugated α-bungarotoxin and for all antibodies used in the IHC studies, we always verified (when receiving a new batch from the manufacturer) that labeling in the spinal cord and/or diaphragm muscle coincided with the established expression pattern of the protein. We have provided Research Resource Identification Initiative (RRID) numbers for all relevant reagents (i.e. antibodies and computer programs) throughout the Materials and methods section.

## Experimental design and statistical analysis

Before starting the study, mice were randomly assigned to experimental groups, and the different vectors used within a given experiment were randomly distributed across these mice (and within a given surgical day). For all the phenotypic analyses, we repeated the experiment for both virus groups in two separate cohorts. All surgical procedures and subsequent behavioral, electrophysiological and histological analyses were conducted in a blinded manner. In the Results section, we provide details of exact n's, group means, standard error of the mean (SEM), statistical tests used and the results of all statistical analyses (including exact p-values, t-values, and F-values) for each experiment and for all statistical comparisons. Statistical significance was assessed by analysis of variance (ANOVA) and multiple comparisons post hoc test. T-test was used for analysis involving only two conditions. Graphpad Prism 6 (Graphpad Software Inc; LaJolla, CA; RRID: SCR_002798) was used to calculate all analyses, and $p < 0.05$ was considered significant.

No samples were omitted from the analysis. All data presented throughout this study represent biological replicates (and not technical replicates). While we included both male and female mice, our analyses are under-powered to examine possible sex-specific effects. Given that males make up a larger portion of the human ALS population, it will be important in followup work to explore the possible sex-specific role of ephrinB2 in ALS pathogenesis.

## Acknowledgements

This work was supported by the Muscular Dystrophy Association (346986 to ACL and MBD; 628389 to DT), the NINDS (R01NS110385 to ACL and MBD; R01NS079702 to ACL; R21NS090912 to DT; RF1AG057882 to DT; R01NS109150 to PP), and the Family Strong for ALS & Farber Family Foundation (PP, DT). Human tissue samples were provided by the NIH NeuroBioBank, Project ALS, and the Jefferson Weinberg ALS Center.

## Additional information

### Funding

| Funder | Grant reference number | Author |
|---|---|---|
| Muscular Dystrophy Association | 346986 | Angelo C Lepore<br>Matthew B Dalva |
| Muscular Dystrophy Association | 628389 | Davide Trotti |
| National Institute of Neurological Disorders and Stroke | R01NS110385 | Angelo C Lepore<br>Matthew B Dalva |
| National Institute of Neurological Disorders and Stroke | R01NS079702 | Angelo C Lepore |
| National Institute of Neurological Disorders and Stroke | R21NS090912 | Davide Trotti |
| National Institute of Neurological Disorders and Stroke | RF1AG057882 | Davide Trotti |
| National Institute of Neurological Disorders and Stroke | R01NS109150 | Piera Pasinelli |
| Family Strong for ALS | | Piera Pasinelli<br>Davide Trotti |
| Farber Family Foundation | | Piera Pasinelli<br>Davide Trotti |

The funders had no role in study design, data collection and interpretation, or the decision to submit the work for publication.

### Author contributions

Mark W Urban, Data curation, Validation, Investigation, Visualization, Methodology, Writing – original draft, Writing – review and editing; Brittany A Charsar, Shashirekha S Markandaiah, Lindsay Sprimont, Wei Zhou, Eric V Brown, Nathan T Henderson, Samantha J Thomas, Biswarup Ghosh, Investigation; Nicolette M Heinsinger, Rachel E Cain, Software, Investigation; Davide Trotti, Piera Pasinelli, Resources, Writing – review and editing; Megan C Wright, Investigation, Writing – review and editing; Matthew B Dalva, Conceptualization, Data curation, Supervision, Funding acquisition, Methodology, Project administration, Writing – review and editing; Angelo C Lepore, Conceptualization, Data curation, Supervision, Funding acquisition, Visualization, Methodology, Writing – original draft, Project administration, Writing – review and editing

### Author ORCIDs

Davide Trotti http://orcid.org/0000-0001-6338-6404
Angelo C Lepore http://orcid.org/0000-0002-5956-6135

### Ethics

All procedures were carried out in compliance with the National Institutes of Health (NIH) Guide for the Care and Use of Laboratory Animals and the ARRIVE (Animal Research: Reporting of In Vivo

Experiments) guidelines. Experimental procedures were approved by Thomas Jefferson University Institutional Animal Care and Use Committee (IACUC). Approved IACUC protocol #01230.

Reviewer #1 (Public Review): https://doi.org/10.7554/eLife.89298.4.sa1
Reviewer #2 (Public Review): https://doi.org/10.7554/eLife.89298.4.sa2
Author Response https://doi.org/10.7554/eLife.89298.4.sa3

---

## Additional files

### Supplementary files
• MDAR checklist

### Data availability
All numerical data generated and analysed during this study are included in the manuscript; source data files have been provided for all six figures.

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
