## [Editor Report · eLife assessment]

This is a **valuable** study of Eph-Ephrin signaling mechanisms generating pathological changes in amyotropic lateral sclerosis. There are exciting findings bearing on the role of glial cells in this pathology. The study emerges with **solid** evidence for a novel astrocyte-mediated mechanism for disease propagation. It may help identify potential therapeutic targets.

---

## [Referee Report · Reviewer #1 (Public Review)]

In the manuscript by Urban et al., the authors attempt to further delineate the role with which non-neuronal CNS cells play in the development of ALS. Towards this goal, the transmembrane signaling molecule ephrinB2 was studied. It was found that there is an increased expression of ephrinB2 in astrocytes within the cervical ventral horn of the spinal cord in a rodent model of ALS. Moreover, reduction of ephrinB2 reduced motoneuron loss and prevented respiratory dysfunction at the NMJ. Further driving the importance of ephrinB2 is an increased expression in the spinal cords of human ALS individuals. Collectively, these findings present compelling evidence implicating ephrinB2 as a contributing factor towards the development of ALS.

---

## [Referee Report · Reviewer #2 (Public Review)]

The contribution of glial cells to the pathogenesis of amyotrophic lateral sclerosis (ALS) is of substantial interest and the investigators have contributed significantly to this emerging field via prior publications. In the present study, authors use a SOD1G93A mouse model to elucidate the role of astrocyte ephrinB2 signaling in ALS disease progression. Erythropoietin-producing human hepatocellular receptors (Ephs) and the Eph receptor-interacting proteins (ephrins) signaling is an important mediators of signaling between neurons and non-neuronal cells in the nervous system. Recent evidence suggests that dysregulated Eph-ephrin signaling in the mature CNS is a feature of neurodegenerative diseases. In the ALS model, upregulated Eph4A expression in motor neurons has been linked to disease pathogenesis. In the present study, authors extend previous findings to a new class of ephrinB2 ligands. Urban et al. hypothesize that upregulated ephrinB2 signaling contributes to disease pathogenesis in ALS mice. The authors successfully test this hypothesis and their results generally support their conclusion.

Major strengths of this work include a robust study design, a well-defined translational model, and complementary biochemical and experimental methods such that correlated findings are followed up by interventional studies. Authors show that ephrinB2 ligand expression is progressively upregulated in the ventral horn of the cervical and lumbar spinal cord through pre-symptomatic to end stages of the disease. This novel association was also observed in lumbar spinal cord samples from post-mortem samples of human ALS donors with a SOD1 mutation. Further, they use a lentiviral approach to drive knock-down of ephrinB2 in the central cervical region of SOD1G93A mice at the pre-symptomatic stage. Interestingly, in spite of using a non-specific promoter, authors note that the lentiviral expression was preferentially driven in astrocytes.

Since respiratory compromise is a leading cause of morbidity in the ALS population, the authors proceed to characterize the impact of ephrinB2 knockdown on diaphragm muscle output. In mice approaching the end stage of the disease, electrophysiological recordings from the diaphragm muscle show that animals in the knock-down group exhibited a ~60% larger amplitude. This functional preservation of diaphragm function was also accompanied with the preservation of diaphragm neuromuscular innervation. However, it must be noted that this cervical ephrinB2 knockdown approach had no impact on disease onset, disease duration, or animal survival. Furthermore, there was no impact of ephrinB2 knockdown on forelimb or hindlimb function. This is an expected result, given the fairly focal approach of ephrinB2 knockdown in C3-C5 spinal segments.

The major limitation of the study is the conclusion that the preservation of diaphragm output following ephrinB2 knockdown in SOD1 mice is mediated primarily (if not entirely) by astrocytes. The authors present convincing evidence that a reduction in ephrinB2 is observed in local astrocytes (~56% transduction) following the intraspinal injection of the lentivirus. However, the proportion of cell types assessed for transduction with the lentivirus in the spinal cord was limited to neurons, astrocytes, and oligodendrocyte lineage cells. Microglia comprise a large proportion of the glial population in the spinal grey matter and have been shown to associate closely with respiratory motor pools. This cell type, amongst the many other that comprise the ventral gray matter, have not been investigated in this study. Nonetheless, there is convincing evidence to suggest astrocytes play a significant role, as compared to oligodendrocytes in promoting ALS pathogenesis.

In summary, this study by Urban et al. provides a valuable framework for Eph-Ephrin signaling mechanisms imposing pathological changes in an ALS mouse model. The role of glial cells in ALS pathology is a very exciting and upcoming field of investigation. The current study proposes a novel astrocyte-mediated mechanism for the propagation of disease that may eventually help to identify potential therapeutic targets.

---

## [Author Response]

The following is the authors’ response to the previous reviews.

**eLife assessment**
This is a valuable study of Eph-Ephrin signaling mechanisms generating pathological changes in amyotropic lateral sclerosis. There are exciting findings bearing on the role of glial cells in this pathology. The study emerges with solid evidence for a novel astrocyte-mediated mechanism for disease propagation. It may help identify potential therapeutic targets.

Response to Editor’s decision letter: Drs. Huang and Zaidi: Thank you for considering this re-revision of our manuscript for potential publication in eLife. We have addressed the remaining comments of reviewer #2. We have included detailed response-to-reviewer comments below to address each of these remaining specific points from reviewer #2, and we have highlighted all the changes in the manuscript text (using a red font color) made in response to these comments. Based on the reviewers’ critiques, we feel our re-working of the manuscript has made for a greatly improved study.

**Reviewer #1 (Recommendations For The Authors):**
Reviewer comment: All questions/concerns have been addressed.

Response: We thank Reviewer #1 for the previous helpful comments that we used to improve our manuscript. As Reviewer #1 has no new comments, we have provided no additional responses to address this reviewer’s input. Instead, we only focus (in this new “Response to Reviewer Comments” document) on the remaining points from Reviewer #2 below.

**Reviewer #2 (Recommendations For The Authors):**
Overall, the authors have addressed most concerns raised in the prior review. A couple of very minor points remain, which would improve the clarity of the report.Reviewer comment 1: The abstract has not been edited and still emphasizes that astrocyte-mediated upregulation in ephrinB2 signaling underlies pathogenicity in mutant SOD1-associated ALS. There is certainly sufficient evidence to suggest a large role for astrocytes, however, without a thorough investigation of other key cell types in the spinal cord, this cannot be concluded specifically.Especially given that a non-specific promoter (U6) was employed in the viral constructs.

Response: We apoplogize for this mistake. In response to the reviewer’s previous comment in the first round of review, we made changes throughout the manuscript to address this issue; however, we failed to do this in the Abstract. In this re-revised manucript, we now also make the necessary changes to the Abstract.

Reviewer comment 2: It is interesting to note that a non-specific promoter, U6, exhibited such large specificity to astrocytes in the cord as compared to neurons (Fig 2M). This is worth discussing briefly in the discussion and how this result compares to those in the literature.

Response: We have now added a brief discussion of this issue to the Discussion section, including describing our previous studies that used the Gfa2 promotor to achieve astrocyte-specific transduction when employing viral vectors in the rodent spinal cord.

Reviewer comment 3: I appreciate the authors including a supplemental figure on the expression of ephrinA4 receptors in the cervical ventral horn. Unfortunately, the quality of this image is very poor in conveying the receptor expression. The detailed discussion point on the expression of EphB receptors in the cervical ventral horn should be sufficient for readers to take into consideration.

Response: We have now removed this supplemental figure and keep only the text from the rerevised manuscript.

Reviewer comment 4: A few instances of motor neuron diameter being attributed to a 200μm2 size remain (e.g. pg 14).

Response: We have corrected this issue throughout the re-revised manuscript. The correct information is: somal diameter greater than 20 μm.

Reviewer comment 5: It is still a little unclear in the result text as to when assessment of lentiviral transduction was conducted following intraspinal injections.

Response: We have now added this detail about the time point of assessing transduction to both the Results section and the Materials/Methods section.

Reviewer comment 6: Some figures are missing markers of significance (e.g. Fig 2M).

Response: Below are our comments about significance markers for each graph in all figures.

Figure 1:

Panel E: We have now added asterisks for any statistically-significant comparisons. In addition, we provide the details of this statistical analysis in the text of the re-revised manuscript.

Figure 2:

Panel M: We have now added asterisks for statistical comparisons, as well as details in the text.

Panel N: The asterisk was already shown in the previous version of the figure.

Figure 3:

Panels B and G: The asterisks were already shown in the previous version of the figure.

Figure 4:

All panels: There are no significant differences; therefore, no asterisks are needed.

Figure 5:

Panel F and G: The asterisks were already shown in the previous version of the figure.

Panel H: The difference is not statistically-signficant.

Figure 6: No graphs are shown in this figure.

Reviewer comment 7: Since a wild type mouse control has not been included in the quantification of diaphragm NMJ innervation with and without ephrin knock-down, it would be useful to include a description or discussion on the phenotype of NMJ denervation exhibited in the SOD1G93A mouse model of ALS.

Response: We have now added description of diaphragm NMJ denervation that occurs inSOD1G93A mice, in particular at the age/time point of our NMJ analysis.